# Bmal1 integrates mitochondrial metabolism and macrophage activation

Ryan K Alexander[1], Yae-Huei Liou[1], Nelson H Knudsen[1], Kyle A Starost[1], Chuanrui Xu[1,2], Alexander L Hyde[1], Sihao Liu[1†], David Jacobi[1‡], Nan-Shih Liao[3], Chih-Hao Lee[1]*

[1]Department of Molecular Metabolism, Division of Biological Sciences, Harvard TH Chan School of Public Health, Boston, United States; [2]School of Pharmacy, Tongji Medical College, Huazhong University of Science and Technology, Wuhan, China; [3]Institute of Molecular Biology, Academia Sinica, Taiwanese, China

*For correspondence:
clee@hsph.harvard.edu

Present address: †Ionis Pharmaceuticals, Carlsbad, United States; ‡l'institut du thorax, INSERM, CNRS, UNIV Nantes & CHU Nantes, Nantes, France

Competing interests: The authors declare that no competing interests exist.

**Abstract** Metabolic pathways and inflammatory processes are under circadian regulation. Rhythmic immune cell recruitment is known to impact infection outcomes, but whether the circadian clock modulates immunometabolism remains unclear. We find that the molecular clock Bmal1 is induced by inflammatory stimulants, including Ifn-γ/lipopolysaccharide (M1) and tumor-conditioned medium, to maintain mitochondrial metabolism under metabolically stressed conditions in mouse macrophages. Upon M1 stimulation, myeloid-specific *Bmal1* knockout (M-BKO) renders macrophages unable to sustain mitochondrial function, enhancing succinate dehydrogenase (SDH)-mediated mitochondrial production of reactive oxygen species as well as Hif-1α-dependent metabolic reprogramming and inflammatory damage. In tumor-associated macrophages, aberrant Hif-1α activation and metabolic dysregulation by M-BKO contribute to an immunosuppressive tumor microenvironment. Consequently, M-BKO increases melanoma tumor burden, whereas administering the SDH inhibitor dimethyl malonate suppresses tumor growth. Therefore, Bmal1 functions as a metabolic checkpoint that integrates macrophage mitochondrial metabolism, redox homeostasis and effector functions. This Bmal1-Hif-1α regulatory loop may provide therapeutic opportunities for inflammatory diseases and immunotherapy.

## Introduction

Inflammation and host defense are energetically costly processes that must balance the use of host resources with an efficient containment of infection or injury. This is underpinned by the dynamic regulation of energy metabolism in immune cells in response to extrinsic signals, including cytokines, pathogen- and damage-associated molecular patterns, and tumor-derived metabolites (*Andrejeva and Rathmell, 2017*; *Buck et al., 2017*; *Ganeshan and Chawla, 2014*; *Hotamisligil, 2017*; *O'Neill et al., 2016*). For instance, activation of macrophages by bacterial products, such as lipopolysaccharide (LPS) from gram-negative bacteria, shifts core metabolic function towards increased reliance on aerobic glycolysis, with concomitant inhibition of mitochondrial respiration (*Fukuzumi et al., 1996*; *Rodríguez-Prados et al., 2010*; *Tannahill et al., 2013*). This depressed mitochondrial function appears to be by design, as this process serves multiple purposes. It leads to the so-called 'broken TCA cycle' that results, in part, from the shunting of citric acid to lipid synthesis (*Andrejeva and Rathmell, 2017*). Itaconate, also derived from citrate/aconitate, can modulate macrophage immune response through different mechanisms (*Lampropoulou et al., 2016*; *Mills et al., 2018*). By contrast, succinate accumulates through anaplerotic reactions, notably glutaminolysis (*Tannahill et al., 2013*). Succinate oxidation to fumarate, mediated by succinate dehydrogenase (SDH)/ETC complex II activity, is a primary source of mitochondrial reactive oxygen species (mROS) in inflammatory macrophages that are involved in bactericidal activity (*Mills et al., 2016*;

*West et al., 2011*). Succinate/SDH is believed to trigger mROS production through accumulation of reduced coenzyme Q leading to reverse electron transfer to ETC complex I (*Chouchani et al., 2014*; *Robb et al., 2018*). These findings demonstrate a well-orchestrated metabolic signaling event that occurs at the expense of reduced fuel economy and compromised mitochondrial function in macrophages.

In addition to their bacteria-killing effect, mROS stabilize hypoxia-inducible factor (Hif)-1α through inhibition of prolyl-hydroxylase enzymes that target Hif-1α for ubiquitination by the von Hippel–Lindau (Vhl) E3 ubiquitin ligase and subsequent proteasomal degradation (*Bell et al., 2007*; *Jaakkola et al., 2001*). Hif-1α is a master transcriptional regulator of genes involved in glycolysis and anabolic metabolism, thereby supplementing the energetic needs of the broken TCA cycle (*Cramer et al., 2003*; *Masson and Ratcliffe, 2014*; *Semenza et al., 1994*). Hif-1α is also required for the expression of the urea cycle enzyme arginase-1 (*Arg1*). Arg1 and nitric oxide synthase 2 were initially designated as markers for M2 and M1 macrophages, as these two enzymes convert the amino acid arginine to citrulline and nitric oxide, respectively. However, M1 activation also upregulates *Arg1* through Hif-1α. Similarly, in the nutrient-deprived tumor microenvironment, tumor-derived lactate has been proposed to increase Hif-1α activity in tumor-associated macrophages (TAMs) and thus to upregulate *Arg1* (*Colegio et al., 2014*). Aberrant expression of Arg1 in TAMs results in local arginine depletion that inhibits antitumor immunity mediated by cytotoxic T cells and natural killer (NK) cells (*Doedens et al., 2010*; *Steggerda et al., 2017*). Accordingly, myeloid-specific deletion of *Hif1a* or *Arg1* suppresses tumor growth in mice (*Colegio et al., 2014*; *Doedens et al., 2010*). These observations suggest that the distinction between M1 and M2 activation may not be as clear in vivo and highlight the importance of energetic regulation in immune cell activation.

The circadian rhythm has been implicated in many biological and pathological processes, including the immune response and tumor progression (*Hardin and Panda, 2013*; *Nguyen et al., 2013*; *Papagiannakopoulos et al., 2016*). The molecular clock includes the master regulator Bmal1 (or Aryl hydrocarbon receptor nuclear translocator-like protein 1, Arntl) and its transcriptional partner Clock, as well as the negative regulatory loop that includes Nr1d1, Nr1d2, period (Per1/2/3) and cryptochrome (Cry1/2) proteins, and the positive regulator loop that includes Rorα/β/γ (*Hardin and Panda, 2013*). Several nuclear receptors, such as the peroxisome proliferator-activated receptors, Pparα, Pparδ/β and Pparγ, are downstream of Bmal1/Clock and control the expression of clock output genes (*Canaple et al., 2006*; *Liu et al., 2013*; *Yang et al., 2006*). The circadian clock is both robust and flexible. It has been demonstrated that time-restricted feeding in mice can synchronize the peripheral clock separately from the central clock (*Damiola et al., 2000*), suggesting that a primary function of circadian rhythm is to maximize metabolic efficiency. In concert, we and others have shown that hepatic Bmal1 regulates rhythmic mitochondrial capacity in anticipation of nutrient availability (*Jacobi et al., 2015*; *Peek et al., 2013*). Prior studies have implicated the circadian oscillator in regulating macrophage inflammatory function. Notably, myeloid-specific *Bmal1* deletion disrupts diurnal monocyte trafficking and increases systemic inflammation and mortality in sepsis mouse models (*Nguyen et al., 2013*). Whether and how the circadian clock controls the metabolism of immune cells to modulate their effector functions remains unclear.

In the present study, we describe a cell-autonomous role for Bmal1 in macrophage energetic regulation. Bmal1 is induced following macrophage inflammatory stimulation. Its loss-of-function exacerbates mitochondrial dysfunction, energetic stress and Hif-1α-dependent metabolic reprogramming. By using the B16-F10 melanoma model, we obtained results that demonstrate that the regulatory axis between Bmal1 and Hif-1α dictates macrophage energy investment that is relevant for discrete activation or polarization states, including activation of M1 and tumor-associated macrophages.

## Results

### The circadian clock is a transcriptional module induced by M1 activation

To assess transcriptional regulators that modulate the energetics and inflammatory function of macrophages, we performed RNA sequencing (RNA-seq) comparing interferon-γ (Ifn-γ) primed bone-marrow-derived macrophages (BMDM) without or with LPS stimulation (10 ng/mL for 8

hr, referred to as M1 activation). Gene ontology analysis using the DAVID platform was performed to identify clusters of transcription factors that were up- or downregulated in inflammatory macrophages, which were used to generate a protein–protein interaction map using STRING (*Table 1* and *Figure 1—figure supplement 1A*). Several activators of mitochondrial function or biogenesis were repressed, including *Myc* (*Li et al., 2005*), *Pparg*, Pparg co-activator 1 beta (*Ppargc1b*), and the mitochondrial transcription factor B1 (*Tfb1m*) and *Tfb2m*. On the other hand, the canonical inflammatory (e.g., *Nfkb1/2*, *Rela/b*, *Hif1a*, interferon regulatory factor 7 (*Irf7*) and *Irf8*) and stress response (e.g., *Atf3*, *Atf6b* and *Nfe2l2*) transcriptional modules were upregulated. Interestingly, clusters of circadian oscillator components (e.g., *Per1*, *Cry1*, *Nr1d1*, *Nr1d2* and *Rora*), as well as nuclear receptors downstream of the molecular clock (e.g., *Ppard* and its heterodimeric partner *Rxra*) (*Liu et al., 2013*), were also induced.

We examined the expression of Bmal1, the non-redundant master regulator of circadian rhythm, and found that M1 activation (*Figure 1A*) or LPS treatment without Ifn-γ priming (*Figure 1B*) induced its mRNA and protein levels, which peaked at 12 hr after the stimulation. Because LPS was directly added to the cell culture without changing the medium, the induction of Bmal1 was not due to serum shock (*Tamaru et al., 2003*). In fact, a one-hour LPS treatment in culture medium with 2% serum was sufficient to reset Bmal1 expression (*Figure 1C*) in a manner resembling serum shock (which requires a much higher serum concentration). Similar results were observed in mouse embryonic fibroblasts (MEFs), suggesting that the inflammatory regulation of Bmal1 was not macrophage-specific (*Figure 1—figure supplement 1B*).

Myeloid-specific *Bmal1* knockout (M-BKO, *Bmal1^{f/f}* crossed to *Lyz2-Cre*) mice were generated to determine the role of the circadian clock in macrophage function. *Bmal1^{f/f}* was used as the wild-type control (WT). M-BKO did not affect M1 induction of canonical inflammatory regulators, such as *Nfkb1*, *Stat3*, *Hif1a* and *Myc* (*Figure 1D*). The expression of genes downstream of Bmal1, including *Nr1d2*, *Cry1* and *Ppard*, was dysregulated, and there was a further reduction of *Pparg* expression by M1 activation in M-BKO macrophages compared to WT cells (*Figure 1D*). By contrast, M2 activation by Il-4 did not regulate *Bmal1* mRNA levels, and Il-4-induced expression of *Arg1* and *Mgl2* was not altered by M-BKO (*Figure 1—figure supplement 1C*). These results suggest that the circadian clock may function as a downstream effector of M1 stimulation in a cell-autonomous manner.

## Bmal1 promotes mitochondrial metabolism in inflammatory macrophages

Because Pparδ/Pparγ are known regulators of mitochondrial function and energy substrate utilization in macrophages (*Dai et al., 2017*; *Kang et al., 2008*; *Lee et al., 2006*; *Odegaard et al., 2007*), we sought to determine the role of Bmal1 in macrophage bioenergetic control. In WT macrophages, M1 activation caused a progressive decrease in mitochondrial content, which was more pronounced in M-BKO macrophages (*Figure 2A*). The reduced mitochondrial content was accompanied by elevated protein levels of the mitophagy receptor Bnip3 (*Figure 2—figure supplement 1A*). The Seahorse Mito Stress test also showed a steeper decline in oxygen consumption rate (OCR) following M1 treatment in M-BKO macrophages, compared to WT cells (*Figure 2B* and *Figure 2—figure supplement 1B*). Measurement of ETC complex activity in isolated mitochondria indicated that M-BKO caused a significant reduction in the activities of complexes II and III, given an equal amount of mitochondrial protein, 6 hr after M1 stimulation (*Figure 2C*). This suggests that macrophage *Bmal1* gene deletion also worsened M1-mediated suppression of mitochondrial function. To determine whether M-BKO affected the basal respiration and/or the recovery of mitochondrial homeostasis following inflammatory insults, we performed Mito Stress tests 24 hr after acute serum shock or LPS treatment, both of which synchronized *Bmal1* gene expression (*Tamaru et al., 2003* and *Figure 1C*). There was no genotypic difference in the OCR at the resting state, and serum shock did not affect respiration in WT and M-BKO macrophages (*Figure 2—figure supplement 1C*). By contrast, the basal OCR of M-BKO macrophages remained suppressed 24 hr following acute LPS treatment, whereas the basal OCR was completely recovered in WT macrophages (*Figure 2—figure supplement 1D*). Thus, *Bmal1* gene deletion impacts macrophage mitochondrial respiration both during inflammatory stimulations and during the subsequent recovery phase.

Seahorse extracellular flux analysis showed that LPS injection increased the extracellular acidification rate (ECAR, indicative of lactic acid secretion) and decreased the oxygen consumption

**Table 1.** Transcriptional regulators that are differentially regulated in M1-activated macrophages.

Genes encoding transcriptional regulators that were significantly induced or repressed by 8h M1 stimulation (p<0.05, false discovery rate [FDR] <0.05, |F.C.| >1.5) in WT bone-marrow-derived macrophages (BMDM) were identified by gene ontology analysis using the DAVID platform. F.C., fold change. Differentially regulated genes that matched the Transcription GO term in the Biological Processes GO database (accession GO:0006350) were used to generate a protein–protein interaction map using String (*Figure 1—figure supplement 1*). Uncharacterized zinc-finger proteins (ZFPs) were omitted from analyses by String.

**Induced (278 genes)**

| | | | | | | | |
|---|---|---|---|---|---|---|---|
| ADAR | CRY1 | GTF2A1 | KDM4B | MNT | PPP1R10 | Snapc1 | Zkscan17 |
| AFF1 | CRY2 | GTF2E2 | KDM5B | MXD1 | PPP1R13L | SNAPC2 | ZMIZ1 |
| AFF4 | CSRNP1 | GTF2F1 | KDM5C | MXI1 | PTOV1 | SOX5 | ZSCAN2 |
| AHR | CSRNP2 | HBP1 | KEAP1 | MYB | PTRF | SPEN | ZSCAN29 |
| AKNA | DAXX | HDAC1 | KLF11 | NAB2 | PURA | SPIC | ZXDB |
| ANP32A | DDIT3 | HES1 | KLF16 | NACC1 | RBPJ | SREBF1 | |
| ARHGAP22 | DDX54 | HES7 | KLF4 | NCOA5 | RCOR2 | SRF | |
| ARID3A | DEDD2 | HEXIM1 | KLF7 | NCOA7 | REL | ST18 | |
| ARID5A | DNMT3A | HIC1 | KLF9 | NCOR2 | RELA | STAT2 | |
| ARNT2 | DPF1 | HIC2 | LCOR | NFAT5 | RELB | STAT3 | |
| ASF1A | DRAP1 | HIF1A | LCORL | NFE2L2 | REST | STAT4 | |
| ATF3 | E2F5 | HIF3A | LHX2 | NFIL3 | RFX1 | STAT5A | |
| ATF4 | E4F1 | HINFP | LIN54 | NFKB1 | RING1 | TAF1C | |
| ATF6B | EAF1 | HIVEP1 | LITAF | NFKB2 | RNF2 | TAF7 | |
| ATXN7L3 | EDF1 | HIVEP2 | LMO4 | NFKBIZ | RORA | TAL1 | |
| BANP | EGR2 | HIVEP3 | MAF | NOTCH1 | RREB1 | TBL1X | |
| BATF | EID3 | HLX | MAFF | NPTXR | RSLCAN18 | TCEB2 | |
| BCL3 | EIF2C1 | HMG20B | MAFG | NR1D1 | RUNX2 | TCF4 | |
| BCL6 | ELK1 | HMGA1 | MAFK | NR1D2 | RUNX3 | TGIF1 | |
| BCORL1 | ELL | HMGA1-RS1 | MAML1 | NR1H2 | RUVBL2 | THAP7 | |
| BHLHE40 | ELL2 | HMGN5 | MAX | NR1H3 | RXRA | TLE2 | |
| BHLHE41 | ELL3 | HOPX | MBD2 | NR2F6 | RYBP | TLE3 | |
| BRWD1 | EPAS1 | HSF4 | MDFIC | NR4A1 | SAFB2 | TRERF1 | |
| BTG2 | ERF | IFI205 | MECP2 | NR4A2 | SAP130 | TRIB3 | |
| CAMTA2 | ERN1 | IFT57 | MED13 | NR4A3 | SAP30 | TRRAP | |
| CASZ1 | ESRRA | ILF3 | MED13L | PAF1 | SBNO2 | TSC22D4 | |
| CBX4 | ETS1 | ING2 | MED15 | PAX4 | SCAF1 | TSHZ1 | |
| CCDC85B | ETV3 | IRF2BP1 | MED25 | PCGF3 | SEC14L2 | USP49 | |
| CDKN2A | FIZ1 | IRF4 | MED26 | PCGF5 | SERTAD1 | VPS72 | |
| CEBPB | FLII | IRF7 | MED28 | PER1 | SETD8 | WHSC1L1 | |
| CEBPD | FOXP1 | IRF8 | MED31 | PER2 | SFPI1 | ZBTB17 | |
| CITED4 | FOXP4 | JARID2 | MEF2D | PHF1 | SIN3B | ZBTB24 | |
| CREB5 | GATA2 | JDP2 | MIER2 | PHF12 | SIX1 | ZBTB46 | |
| CREBBP | GATAD2A | JMJD6 | MIER3 | PIAS4 | SIX5 | ZBTB7A | |
| CREBL2 | GATAD2B | JUN | MITF | PML | SLC30A9 | ZBTB7B | |
| CREBZF | GFI1 | JUNB | MIXL1 | POU2F2 | SMAD3 | ZEB1 | |
| CREM | GLIS3 | JUND | MKL1 | POU3F1 | SMAD4 | ZFHX4 | |
| CRTC2 | GPBP1 | KDM3A | MLL1 | POU6F1 | SMAD7 | ZGPAT | |
| CRTC3 | GRHL1 | KDM4A | MNDA | PPARD | SMYD1 | ZHX2 | |

*Table 1 continued on next page*

**Repressed** (195 genes)

| | | | | |
|---|---|---|---|---|
| ACTL6A | ELK3 | IRF2 | NAA15 | SAP18 |
| AHRR | ELP2 | ITGB3BP | NCOA1 | SAP25 |
| AI987944 | ELP3 | KDM2B | NCOA3 | SETD7 |
| ANG | ELP4 | KLF10 | NFATC1 | SETDB1 |
| ASCC1 | ENY2 | KLF13 | NFATC2 | SNAPC5 |
| ASF1B | ERCC8 | KLF2 | NFIA | SP3 |
| ATAD2 | ESR1 | KLF8 | NKRF | SSBP2 |
| AW146154 | ETOHI1 | L3MBTL2 | NPAT | SSRP1 |
| BCL9L | ETV1 | LBH | NPM3 | STAT1 |
| CBFA2T3 | EYA1 | LRPPRC | NR2C1 | SUV39H1 |
| CBX3 | EYA4 | LYL1 | NRIP1 | SUV39H2 |
| CBX6 | EZH2 | MAFB | OVOL2 | SUV420H2 |
| CBX8 | FLI1 | MARS | PA2G4 | TADA1 |
| CCNH | FNTB | MBTPS2 | PHF19 | TADA2A |
| CDCA7 | FOXM1 | MCM2 | PHTF2 | TAF4B |
| CDCA7L | GTF2H2 | MCM3 | PNRC2 | TAF9B |
| CEBPA | GTF2I | MCM4 | POLR1B | TBX6 |
| CEBPG | GTF2IRD1 | MCM5 | POLR2G | TCEA3 |
| CEBPZ | GTF3A | MCM6 | POLR2I | TCEAL8 |
| CHAF1A | GTF3C5 | MCM7 | POLR3B | TCF7L2 |
| CHAF1B | HABP4 | MCM8 | POLR3H | TFB1M |
| CHD9 | HDAC10 | MCTS1 | POLR3K | TFB2M |
| CHURC1 | HDAC11 | MED14 | PPARG | TFDP2 |
| CIITA | HDAC2 | MED18 | PPARGC1B | THOC1 |
| CIR1 | HDAC6 | MED22 | PRIM1 | TLE1 |
| CREB3 | HDAC7 | MED27 | PRIM2 | TRAPPC2 |
| CREB3L1 | HDAC8 | MEF2A | PRMT7 | TRIM24 |
| CREB3L2 | HDAC9 | MEF2C | PROX2 | TWISTNB |
| CTNND1 | HELLS | MEIS1 | PSPC1 | TXNIP |
| CUX1 | HHEX | MLF1IP | RAD54B | UHRF1 |
| DDI2 | HIP1 | MLL3 | RB1 | USF1 |
| DNMT1 | HIRA | MLLT3 | RBAK | VGLL4 |
| DR1 | HMBOX1 | MNAT1 | RCBTB1 | VPS36 |
| E2F1 | HMGA2 | MPV17 | RCOR3 | WTIP |
| E2F2 | HOXA1 | MXD3 | RERE | ZBTB3 |
| E2F6 | HTATSF1 | MXD4 | RFC1 | ZBTB8A |
| E2F7 | IKBKAP | MYBL2 | RPAP1 | ZHX1 |
| E2F8 | IKZF2 | MYC | RSC1A1 | ZIK1 |
| EGR3 | IL16 | MYCBP2 | RSL1 | ZKSCAN4 |

rate (OCR) of WT macrophages, as expected from aerobic glycolysis (*Figure 2D*). The ECAR and OCR were further enhanced and suppressed, respectively, in M-BKO macrophages. Similar results were obtained in thioglycollate-elicited peritoneal macrophages isolated from WT and M-BKO mice (*Figure 2—figure supplement 1E*). By contrast, stable overexpression of *Bmal1* (Bmal1-OE) in RAW264.7 macrophages resulted in higher OCR and lower ECAR after LPS

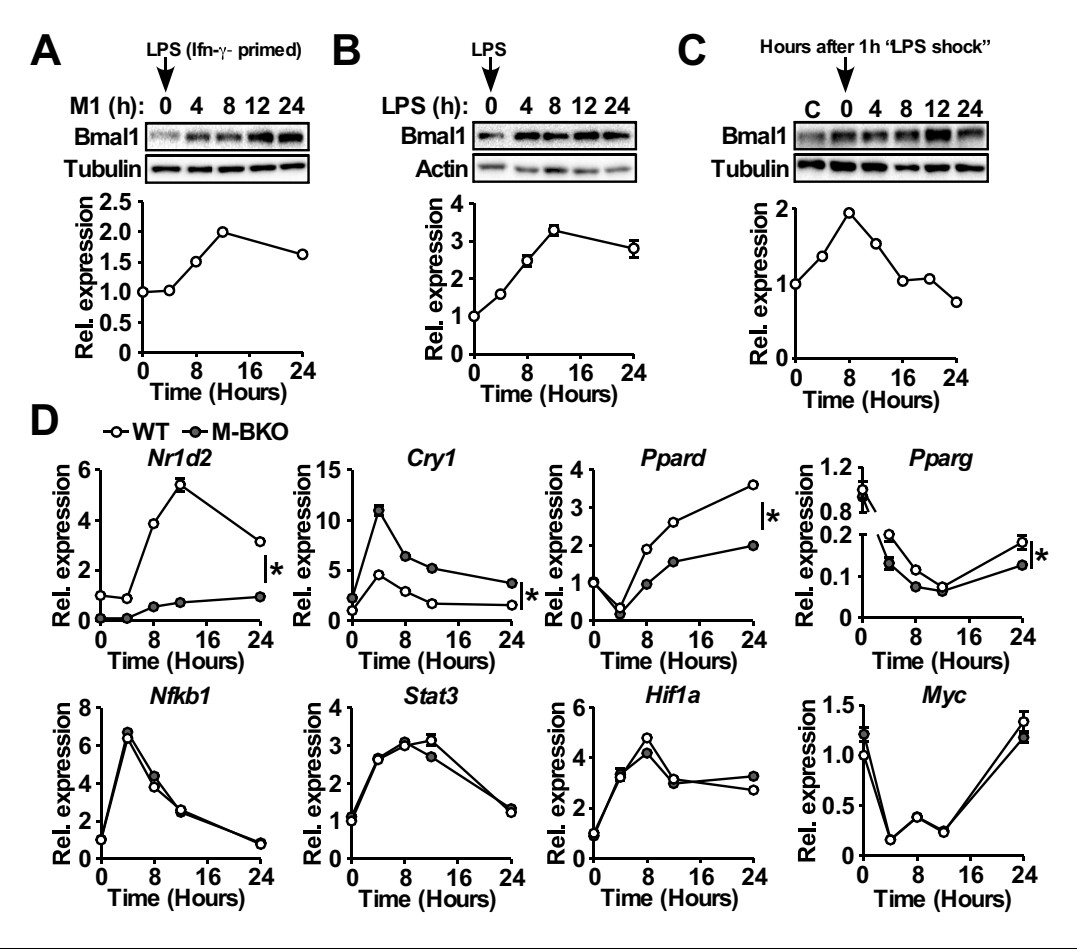

**Figure 1.** Macrophage Bmal1 is induced by M1 activation. (A–C) Bmal1 protein levels (top) and relative gene expression determined by qPCR (bottom) in bone-marrow-derived macrophages (BMDM) during a 24-hr time course of M1 activation (10 ng/ml Ifn-γ overnight priming + 10 ng/ml LPS) (A), treatment with LPS alone (100 ng/ml) (B), or acute LPS treatment for 1 hr (100 ng/mL) (C). For M1- and LPS-only treatments, LPS was spiked in at time zero without medium change. For acute LPS treatment, cells were grown with LPS for one hour followed by culture in DMEM, 2% FBS without LPS (time zero indicates medium change). N = 3 biological replicates were used for qPCR. (D) Relative expression of circadian clock and inflammatory transcriptional regulators in M1-activated macrophages, as determined by qPCR. N = 3 biological replicates, statistical analysis performed using two-way ANOVA for WT vs M-BKO across the time course. Data are presented as mean ± S.E.M. *, p<0.05. Experiments were repeated at least twice. The online version of this article includes the following figure supplement(s) for figure 1:

**Figure supplement 1.** The circadian clock is a transcriptional module that is induced by M1 activation.

stimulation, compared to control cells (*Figure 2—figure supplement 1F-G*). To examine aerobic glycolysis directly, glucose was injected during the extracellular flux assay with or without co-injection of LPS. There was no difference in the basal glycolytic rate between WT and M-BKO macrophages (*Figure 2E*). LPS increased ECAR in both genotypes and to a greater extent in M-BKO macrophages. The induced ECAR could be blocked by injection of 2-deoxyglucose (2-DG), confirming that the acidification was caused by aerobic glycolysis. Furthermore, an increase in the glycolytic rate was observed in splenic macrophages from M-BKO mice isolated 6 hr after i.p. injection of LPS, which was accompanied by lowered circulating glucose levels, indicative of increased glucose consumption by inflammatory myeloid cells in M-BKO mice when compared to WT animals (*Figure 2—figure supplement 1H-I*).

To further assess the metabolic state, metabolomics analyses were employed to compare the cellular metabolite levels of WT and M-BKO macrophages 0, 6 and 12 hr after M1 activation

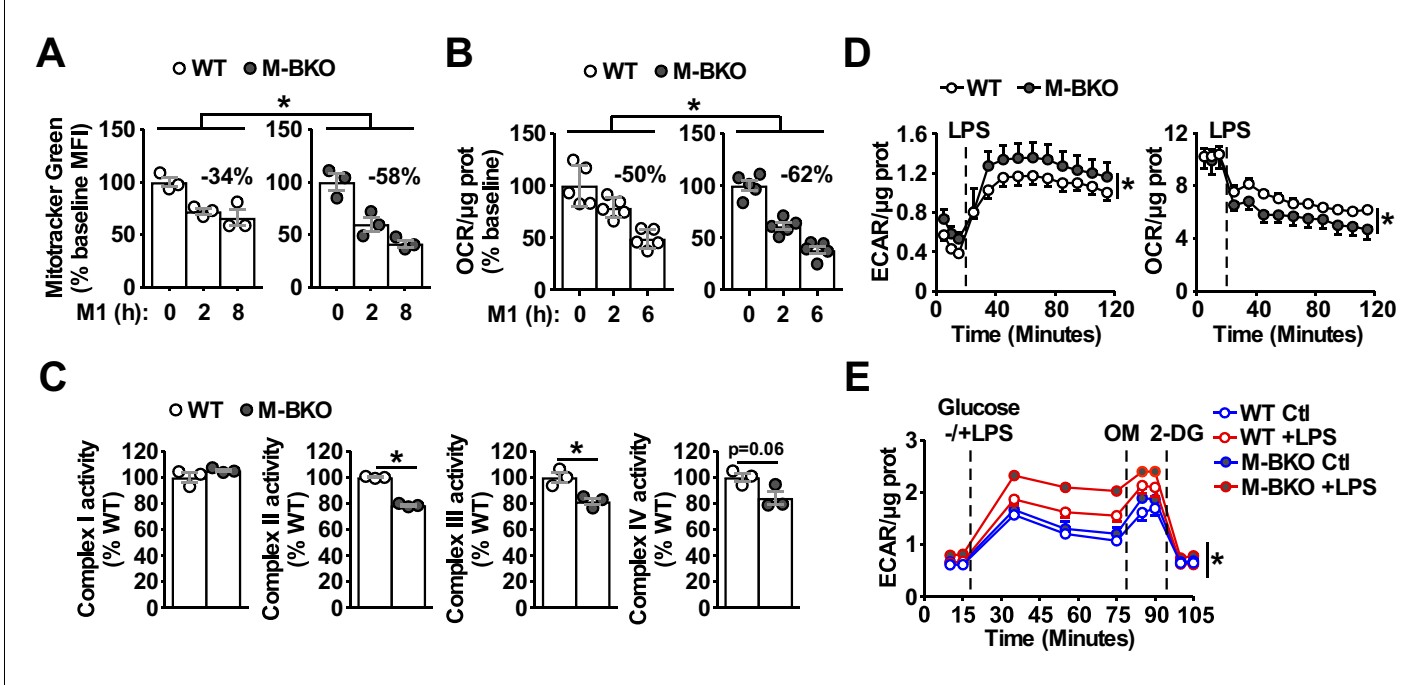

**Figure 2.** Bmal1 is required to maintain mitochondrial metabolism. (**A**) Assessment of mitochondrial mass in macrophages throughout a time course of M1 activation using Mitotracker Green (mean fluorescence intensity, MFI) determined by flow cytometry. N = 3 biological replicates, statistical analysis was performed using two-way ANOVA for WT vs M-BKO across the time course. (**B**) Basal oxygen consumption rate (OCR) in Ifn-γ-primed macrophages pretreated without or with LPS (10 ng/mL) for 2 or 6 hr before the assay. See *Figure 2—figure supplement 1B* for the full assay results. The assay medium contained minimal DMEM with 5 mM glucose and 1 mM sodium pyruvate, pH 7.4. N = 5 biological replicates, statistical analysis was performed using two-way ANOVA for WT vs M-BKO across the time course. (**C**) Activities of ETC complexes in isolated mitochondria from WT and M-BKO macrophages after 6 hours M1 stimulation. N = 3 biological replicates, statistical analysis was performed using Student's T test. (**D**) Extracellular flux analysis in Ifn-γ-primed macrophages measuring the changes in extracellular acidification rate (ECAR, left panel) and oxygen consumption rate (OCR, right panel) following LPS injection (100 ng/mL). The assay medium contained 5 mM glucose and 1 mM pyruvate in minimal DMEM with 2% dialyzed FBS, pH 7.4. N = 5 biological replicates, statistical analysis was performed using two-way ANOVA for WT vs M-BKO across the time course. (**E**) Glycolytic stress test in Ifn-γ-primed macrophages measuring ECAR following glucose (25 mM) injection, with or without LPS (100 ng/mL). Maximal glycolytic rate was determined by injection of oligomycin (OM, 2 µM), and glycolysis-dependent ECAR was determined by injection of 2-deoxyglucose (2-DG, 50 mM). The assay medium contained minimal DMEM with 2% dialyzed FBS, pH 7.4. N = 5 biological replicates, statistical analysis was performed using two-way ANOVA for WT vs M-BKO across the time course. Data are presented as mean ± S.E.M. *, p<0.05. Experiments were repeated at least twice.

The online version of this article includes the following figure supplement(s) for figure 2:

**Figure supplement 1.** Effects of *Bmal1* gene deletion and overexpression on glycolytic versus oxidative metabolism.

(*Figure 3A–B* and *Figure 3—source data 1*). As has been reported (*Tannahill et al., 2013*), M1 activation caused accumulation of glycolytic intermediates (glucose-6-phosphate, fructose-6-phosphate and lactic acid) and depletion of TCA metabolites (e.g. citrate) but accumulation of succinate. Glycolytic metabolites and succinate were significantly higher in M-BKO macrophages than in WT cells. M-BKO cells also showed accumulation of several amino acids and intermediates of the urea cycle (which detoxifies ammonia released from amino-acid deamination) (*Figure 3A* and *Figure 3—source data 1*). Consistent with the increased glycolytic metabolites, M1-stimulated glucose uptake and lactate production were higher in M-BKO macrophages than in WT cells (*Figure 3C–D*). These results suggest that *Bmal1* loss-of-function leads to metabolic dysregulation in M1-stimulated macrophages.

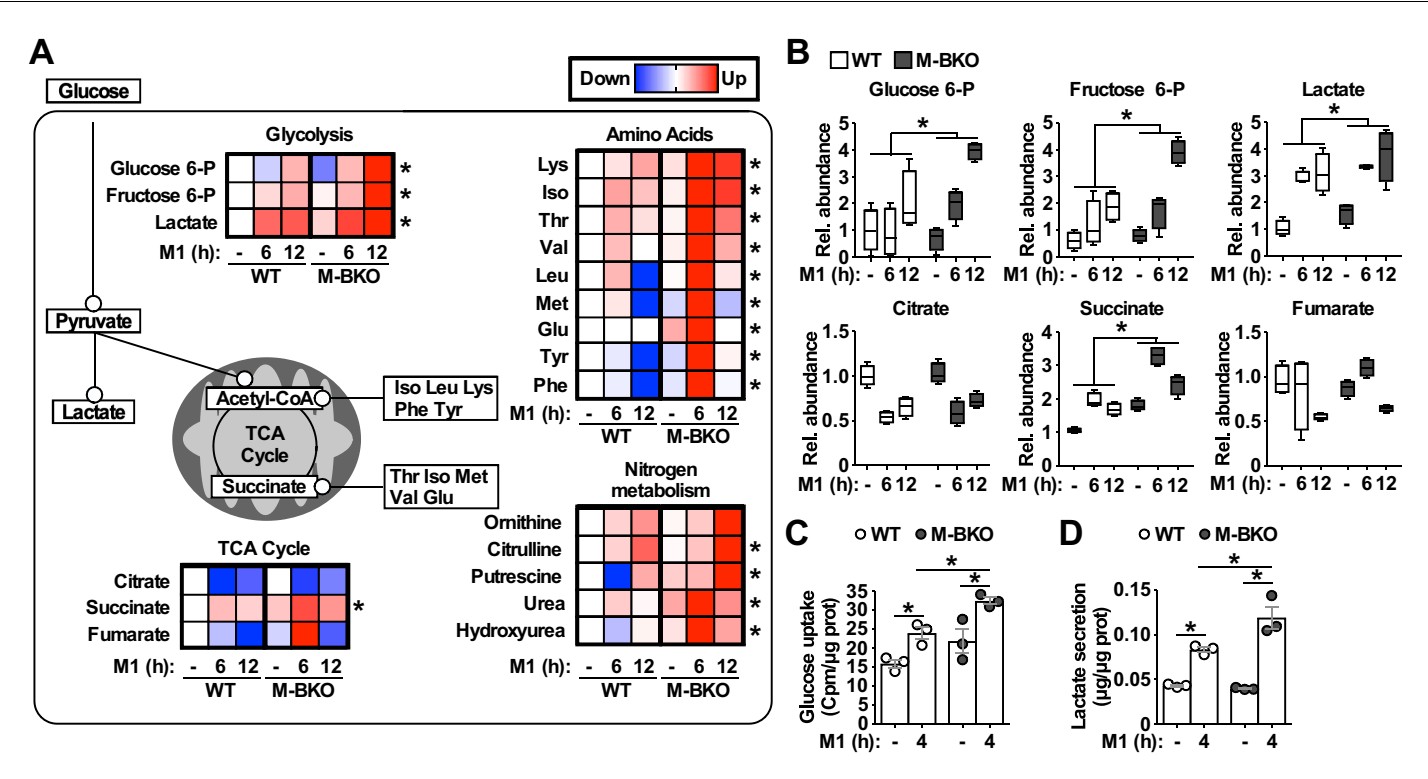

**Figure 3.** *Bmal1* deletion induces a metabolic shift for glycolytic and amino-acid metabolism. (**A**) Summary of steady-state metabolomics data for differentially regulated metabolites from WT and M-BKO macrophages throughout a 12-hour M1 activation time course. Data are presented as heat maps (normalized to WT control for each metabolite, each panel is the average of four biological replicates). Statistical analysis was performed using two-way ANOVA for WT vs M-BKO across the time course. (**B**) Box plots of the relative abundances of select metabolites in panel (**A**). (**C, D**) Uptake of [³H]—2-deoxyglucose (**C**) and lactate secretion (**D**) in control or M1-activated macrophages. N = 3 biological replicates, statistical analysis was performed using Student's T test. Cell culture assays were repeated at least twice.

The online version of this article includes the following source data for figure 3:

**Source data 1.** Metabolomics data for M1-activated WT and M-BKO macrophages.

## Bmal1–Hif-1α crosstalk regulates macrophage energy metabolism

As mentioned earlier, Hif-1α is a primary regulator of glucose metabolism in inflammatory macrophages. The enhanced aerobic glycolysis in M-BKO macrophages prompted us to examine whether Hif-1α activity was aberrantly elevated in these cells. Western blot analyses revealed that M1 activation led to a several-fold induction of Hif-1α protein levels in M-BKO macrophages compared to WT cells (*Figure 4A*), whereas Bmal1-OE RAW264.7 macrophages showed reduced Hif-1α protein (*Figure 4—figure supplement 1A*). The expression of Hif-1α targets, such as lactate dehydrogenase A (*Ldha*), *Arg1* and *Il1b*, was enhanced by M-BKO and blocked by myeloid *Hif1a* knockout (M-HKO, *Figure 4B*). Hif-1α gene expression did not differ between WT and M-BKO cells (*Figure 1D*). mROS derived from increased succinate oxidation has previously been demonstrated to stabilize Hif-1α protein in inflammatory macrophages (*Mills et al., 2016*). Metabolite analyses showed accumulation of succinate in M-BKO macrophages, suggesting that elevated mROS may be the cause of the increased Hif-1α protein. In fact, levels of mROS were higher in isolated mitochondria from M-BKO macrophages at 1 hr and 4 hr of M1 activation than in WT macrophages (*Figure 4C*). The addition of succinate increased mROS production in mitochondria from both WT and M-BKO macrophages. An additional two-fold induction of mROS was detected in mitochondria from 4-hr M1-stimulated M-BKO, but not in those from WT macrophages. Hif-1α protein accumulation could be normalized between genotypes by co-treatment with the antioxidant N-acetylcysteine (N-AC) or the competitive complex II inhibitor dimethylmalonate (DMM), which blocks mROS production (*Figure 4D*).

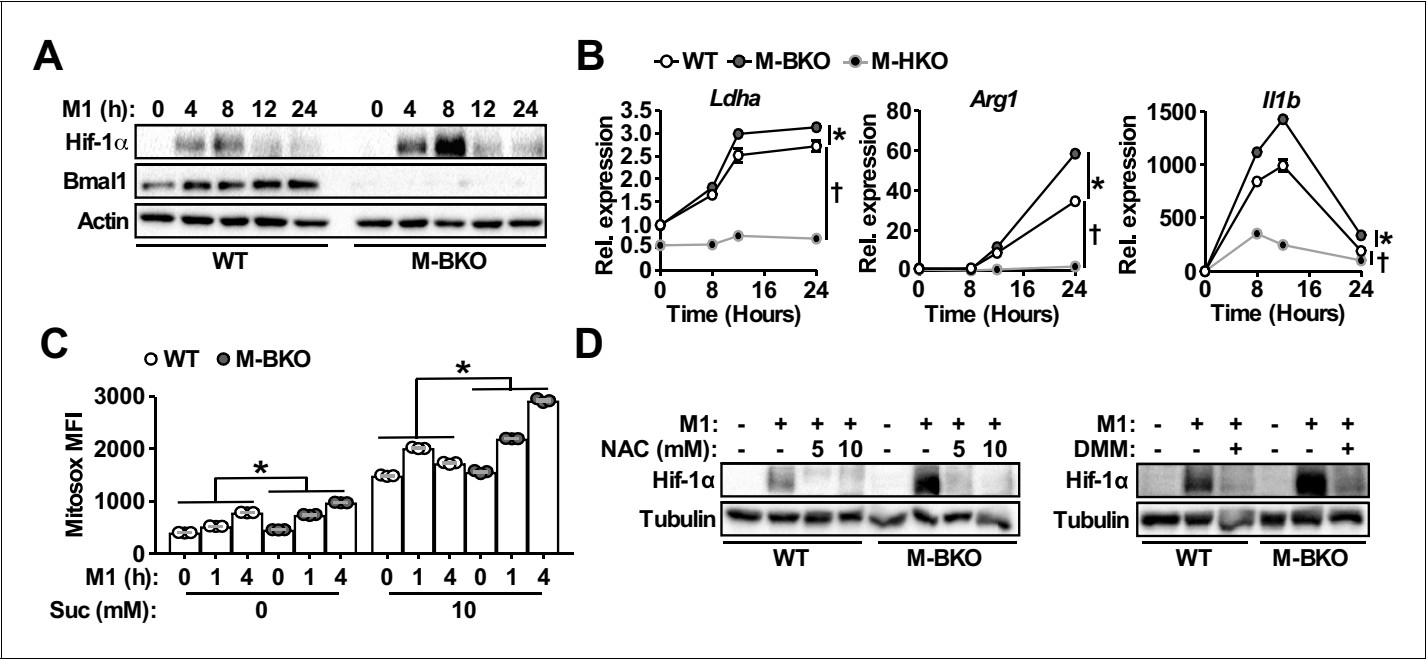

**Figure 4.** *Bmal1* loss-of-function increases oxidative stress and Hif-1α protein accumulation. (**A**) Immunoblots of Hif-1α and Bmal1 protein levels in WT and M-BKO macrophages during a 24-hr time course of M1 activation. (**B**) Relative expression of Hif-1α target genes in WT, M-BKO and M-HKO macrophages determined by qPCR. N = 3 biological replicates, statistical analysis was performed using two-way ANOVA for WT vs M-BKO or WT vs M-HKO across the time course. (**C**) Measurement of mROS using MitoSox Red (mean fluorescence intensity, MFI) in mitochondria isolated from control or M1-activated macrophages. Succinate (Suc, 10 mM) was included during MitoSox Red staining where indicated. N = 3 biological replicates, statistical analysis was performed using two-way ANOVA for WT vs M-BKO across the time course. (**D**) Hif-1α protein levels in control or 8-hr M1-activated macrophages co-treated with or without N-acetylcysteine (NAC) or 10 mM dimethylmalonate (DMM). Data are presented as mean ± S.E.M. *, p<0.05 for WT vs M-BKO and p<0.05 for WT vs M-HKO. Experiments were repeated at least twice.

The online version of this article includes the following figure supplement(s) for figure 4:

**Figure supplement 1.** Increased oxidative stress and Hif-1α activity in M1 activated M-BKO macrophages.

Furthermore, M1-stimulated glucose uptake, lactate release and aerobic glycolysis were attenuated in myeloid-specific *Bmal1* and *Hif1a* double knockout macrophages (M-BHdKO; *Figure 4—figure supplement 1B-D*), indicating that the increased glucose utilization in M-BKO was Hif-1α-dependent.

A previous study suggests that *Bmal1* deletion impairs the expression of *Nfe2l2* (which encodes Nrf2) and its downstream antioxidant genes, thereby increasing oxidative stress (*Early et al., 2018*). However, we found that expression of *Nfe2l2*- and Nrf2-induced oxidative stress responsive genes, such as NAD(P)H quinone dehydrogenase 1 (*Nqo1*; *Figure 4—figure supplement 1E*), were upregulated in M-BKO macrophages upon M1 stimulation, suggesting that increased mROS associated with M-BKO was the cause rather than the consequence of dysregulated Nrf2 signaling. Collectively, these data indicate that Bmal1 and Hif-1α regulate opposing metabolic programs and that Bmal1-mediated mitochondrial metabolism serves to fine-tune Hif-1α activity by modulating oxidative stress.

## *Bmal1* loss-of-function induces metabolic reprogramming toward amino-acid catabolism

To characterize fully metabolic programs that were impacted by *Bmal1* loss of function, we compared RNA-seq data from control and M1-activated WT and M-BKO macrophages. These analyses revealed that the majority of M1-induced or -suppressed genes were regulated in a similar manner in WT and M-BKO macrophages, suggesting that *Bmal1* gene deletion affected specific inflammatory processes (*Figure 5—figure supplement 1A* and *Figure 5—source data 1*) and in line with the intact expression pattern of Nfkb1 in M-BKO macrophages (*Figure 1D*). Gene ontology

analyses indicated that the most enriched categories of M1-upregulated genes shared by both genotypes included regulation of apoptosis, response to stress and cytokine production. Among the top categories of suppressed genes were the cell cycle, DNA repair and carbohydrate metabolism. In the carbohydrate metabolism category, most TCA cycle enzymes were downregulated by M1 activation (*Figure 5A*).

Direct comparison between M1-stimulated WT and M-BKO macrophages identified 419 genes that are more highly expressed in M1-activated M-BKO macrophages (FDR < 0.05, p<0.05, *Figure 5—figure supplement 1B* and *Figure 5—source data 1*). Most-enriched categories included stress and inflammatory responses that contained *Il1b* and other Hif-1α target genes, such as *S100a8/S100a9* (*Grebhardt et al., 2012*). Other top enriched pathways were protein catabolism and amino-acid transport. These pathways included genes encoding plasma membrane amino-acid transporters (e.g., *Slc7a2*, *Slc7a8*, *Slc7a11*, *Slc38a2* and *Slc38a7*) as well as ubiquitin-activating, -conjugating and -ligating enzymes that target proteins for proteasomal degradation (e.g., ubiquitin-like modifier-activating enzyme 6 [*Uba6*], ubiquitin conjugating enzymes [*Ube2q2* and *Ube2e3*], ring finger proteins [*Rnf12*, *Rnf56*, *Rnf128*, and *Rnf171*], cullin 3 [*Cul3*] and *Cul5*, and ubiquitin protein ligase e3a [*Ube3a*]) (*Figure 5A–B*). The expression of enzymes that are involved in the breakdown of branched-chain amino acids was also higher in M1-activated M-BKO cells, including branched-chain keto acid dehydrogenase E1 subunit beta (*Bckdhb*) and methylmalonate semialdehyde dehydrogenase (*Mmsdh*). These results are consistent with the increased amino-acid catabolism that was observed in metabolite assays (*Figure 3A*). Interestingly, certain genes described above, notably *Slc7a8*, appeared to be counter-regulated by Hif-1α, as their induction by M1 stimulation was blunted in M-HKO macrophages (*Figure 5—figure supplement 1C*).

Slc7a8, also called L-type amino-acid transporter 2 (Lat2), transports neutral amino acids that could be converted to succinate and could potentially contribute to Hif-1α protein stabilization. In line with increased amino acid-metabolism, extracellular flux analysis showed that M-BKO macrophages showed enhanced glutamine utilization compared to WT cells, which was blocked by 2-amino-bicyclo-(2,2,1)-heptane-2-carboxylate (BCH), an L-type amino-acid transporter inhibitor (*Christensen et al., 1969*; *Segawa et al., 1999*; *Figure 5—figure supplement 1D*). BCH decreased and normalized levels of Hif-1α protein between WT and M-BKO macrophages (*Figure 5C*). In addition, treatment with either BCH or DMM suppressed the expression of *Il1b*, *Slc7a8* and *Slc7a11* induced by M1 stimulation (*Figure 5D*). The combination of BCH and DMM did not exert a greater effect over that of DMM alone. Thus, amino-acid metabolism is upregulated in response to dysregulated energy metabolism in M-BKO macrophages, which contributes to increased oxidative stress and Hif-1α activation.

## Macrophage *Bmal1* gene deletion promotes an immune-suppressive tumor-associated macrophage phenotype and enhances tumor growth

It has been suggested that myeloid-specific *Bmal1* deletion disrupts diurnal monocyte trafficking, thereby increasing sepsis-induced systemic inflammation and mortality (*Nguyen et al., 2013*). Our results suggest that the cell-autonomous function of Bmal1 on macrophage metabolism and Hif-1α activation may contribute to the reported phenotype. Hif-1α regulates the polarization of M1 and tumor-associated macrophages, both of which are under energetically challenged conditions. We sought to determine whether Bmal1–Hif-1α crosstalk plays a role in modulating TAM activation through a mechanism similar to that in M1 stimulation. Treatment of macrophages with conditioned medium from primary B16-F10 tumors (T-CM) increased the expression of both *Bmal1* mRNA and Bmal1 protein (*Figure 6A*). When compared to WT macrophages, M-BKO macrophages showed enhanced mROS production and Hif-1α protein induced by T-CM (*Figure 6B–C*). Tracking with Hif-1α stabilization, aerobic glycolysis was upregulated by T-CM pretreatment in WT and to a greater extent in M-BKO macrophages (*Figure 6D*). T-CM elicited an energetic stress gene expression signature resembling that of M1 stimulation, which included upregulation of amino acid metabolism (*Arg1*, *Slc7a8* and *Bckdhb*) and oxidative stress (*Slc7a11* and *Nqo1*) pathways in WT macrophages that were further induced by M-BKO (*Figure 6E*).

Subsequently, we employed a mouse model of melanoma through subcutaneous injection of B16-F10 melanoma cells to assess the impact of myeloid *Bmal1* deletion on tumor growth. Tumor volume was increased in both male and female M-BKO mice compared to WT controls (*Figure 6F*). Furthermore, the expression of *Arg1*, *Slc7a8* and *Slc7a11* was upregulated in F4/80+ cells isolated

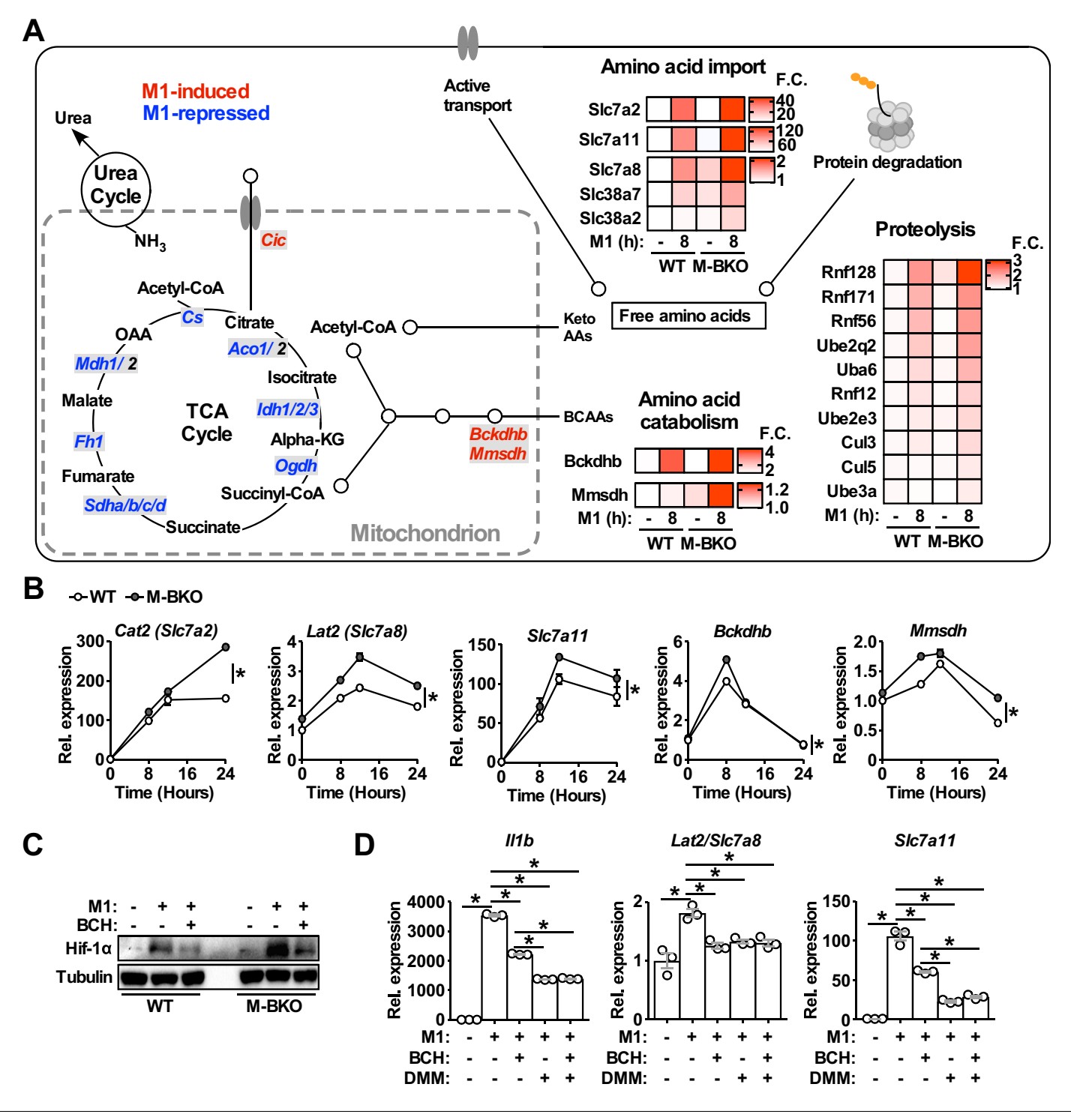

**Figure 5.** Genes involved in amino-acid uptake and catabolism are upregulated in M-BKO macrophages. (A) Schematic representation of M1-regulated genes involved in amino-acid and TCA metabolism determined by RNA-seq. Genes in blue are downregulated whereas genes in red are upregulated by 8 hours M1 activation in both WT and M-BKO macrophages. Genes that are differentially regulated in these two genotypes are displayed in heat maps on the right. F.C., fold change. N = 3 biological replicates. BCAAs, branch chain amino acids; Keto AAs, ketogenic amino acids; Cic, mitochondrial citrate carrier. (B) Relative expression of differentially regulated genes identified by RNA-seq and validated by qPCR in a 24-hr time course of M1 activation. N = 3 biological replicates, statistical analysis was performed using two-way ANOVA for WT vs M-BKO across the time course. (C) Hif-1α protein levels in control or 6-hr M1-activated macrophages with or without co-treatment of the neutral amino-acid transport inhibitor 2-amino-2-norbornanecarboxylic acid (BCH, 10 mM). (D) Gene expression in control or 6-hr M1-activated macrophages with or without co-treatment of 10

*Figure 5 continued on next page*

*Figure 5 continued*

mM BCH and/or the complex II inhibitor dimethyl malonate (DMM, 10 mM) determined by qPCR. N = 3 biological replicates, statistical analysis was performed using Student's T test. Data are presented as mean ± S.E.M. *, p<0.05. Cell culture experiments were repeated at least twice.

The online version of this article includes the following source data and figure supplement(s) for figure 5:

**Source data 1.** Functional annotation clustering genes commonly and differentially regulated between WT and M-BKO BMDMs by 8 h M1 activation.
**Figure supplement 1.** Transcriptome analysis of genes that are regulated by M1 stimulation in WT and M-BKO macrophages.

from tumors but not spleens of M-BKO mice compared to WT animals (*Figure 6G*). Of note, the mRNA levels of *Arg1*, *Slc7a8* and *Slc7a11* were substantially higher in tumor than in splenic F4/80[+] cells. Flow cytometry analyses of F4/80[+] cells from primary tumors stained with Mitosox Red and 2-deoxy-2-[(7-nitro-2,1,3-benzoxadiazol-4-yl)amino]-D-glucose (2-NBDG) were employed to assess mROS production and glucose uptake, respectively. M-BKO TAMs exhibited a trend towards increased mROS and significantly higher glucose uptake, compared to WT TAMs (*Figure 6—figure supplement 1A-C*). These results demonstrate that the metabolic reprogramming observed in T-CM-primed BMDM is shared by TAMs.

To confirm that macrophage Bmal1 modulates tumor growth cell-autonomously and to assess the effect of TAMs on anti-tumor immune response within the same host environment, we co-injected B16-F10 cells with either WT or M-BKO macrophages into the right or left flanks, respectively, of WT mice. Tumor growth rate was substantially higher when the tumor cells were co-injected with M-BKO macrophages than when co-injected with WT cells (*Figure 7A*). In concert, co-injection with M-BKO macrophages led to a reduction in the CD8[+] T cell population among tumor-infiltrating CD45[+] leukocytes, as well as to functionally primed CD8[+] T and NK cells that expressed Ifn-γ protein following stimulation with phorbol myristate acetate and ionomycin ex vivo (*Figure 7B*). Similar results were obtained when the co-injections were performed in M-BKO mice (*Figure 7—figure supplement 1A, B*).

We next sought to address the importance of oxidative stress in TAM activation. Similar to M1 macrophages, DMM blocked Hif-1α protein accumulation and attenuated *Arg1* upregulation in T-CM-treated macrophages (*Figure 7—figure supplement 1C-D*). Administering DMM (~150 mg/kg body/day) at the time of macrophage-tumor cell co-inoculation effectively suppressed melanoma tumor growth and normalized the difference in tumor promoting effects between WT and M-BKO macrophages (*Figure 7C*). These results reveal a unifying mechanism by which Bmal1 controls macrophage effector functions through bioenergetic regulation, and suggest that targeting oxidative stress may provide a means to modulate the anti-tumor activity of TAMs.

## Discussion

It has been reported that sepsis exerts a long-lasting effect on circadian rhythm alteration in mice (*Marpegán et al., 2005*; *O'Callaghan et al., 2012*). In the current study, we show that inflammatory stimulants, including Ifn-γ/LPS and tumor-derived factors, control the expression of the circadian master regulator Bmal1 in the macrophages. Our data further demonstrate that Bmal1 is an integral part of the metabolic regulatory network and modulates macrophage activation, in part through crosstalk with Hif-1α. The Bmal1–Hif-1α regulatory loop regulates the balance between oxidative and glycolytic metabolism in energetically stressed macrophages that have distinct effector functions. *Bmal1* loss-of-function in M1-activated macrophages causes mitochondrial dysfunction, thereby potentiating mROS production and Hif-1α protein stabilization, which probably contributes to the increased sepsis-induced inflammatory damage reported for M-BKO mice (*Nguyen et al., 2013*). Within the tumor microenvironment, macrophage *Bmal1* gene deletion leads to compromised anti-tumor immunity and accelerated tumor growth in a mouse melanoma model. Therefore, the Bmal1–Hif-1α nexus serves as a metabolic switch that may be targeted to control macrophage effector functions.

Much attention has been focused on how inflammatory stimuli disrupt mitochondrial metabolism as a means to generate signaling molecules, including TCA metabolites and mROS. The analysis of transcriptional modules that are involved in macrophage inflammatory response reveals a coordinated effort in the control of mitochondrial activity. The expression of several regulators of

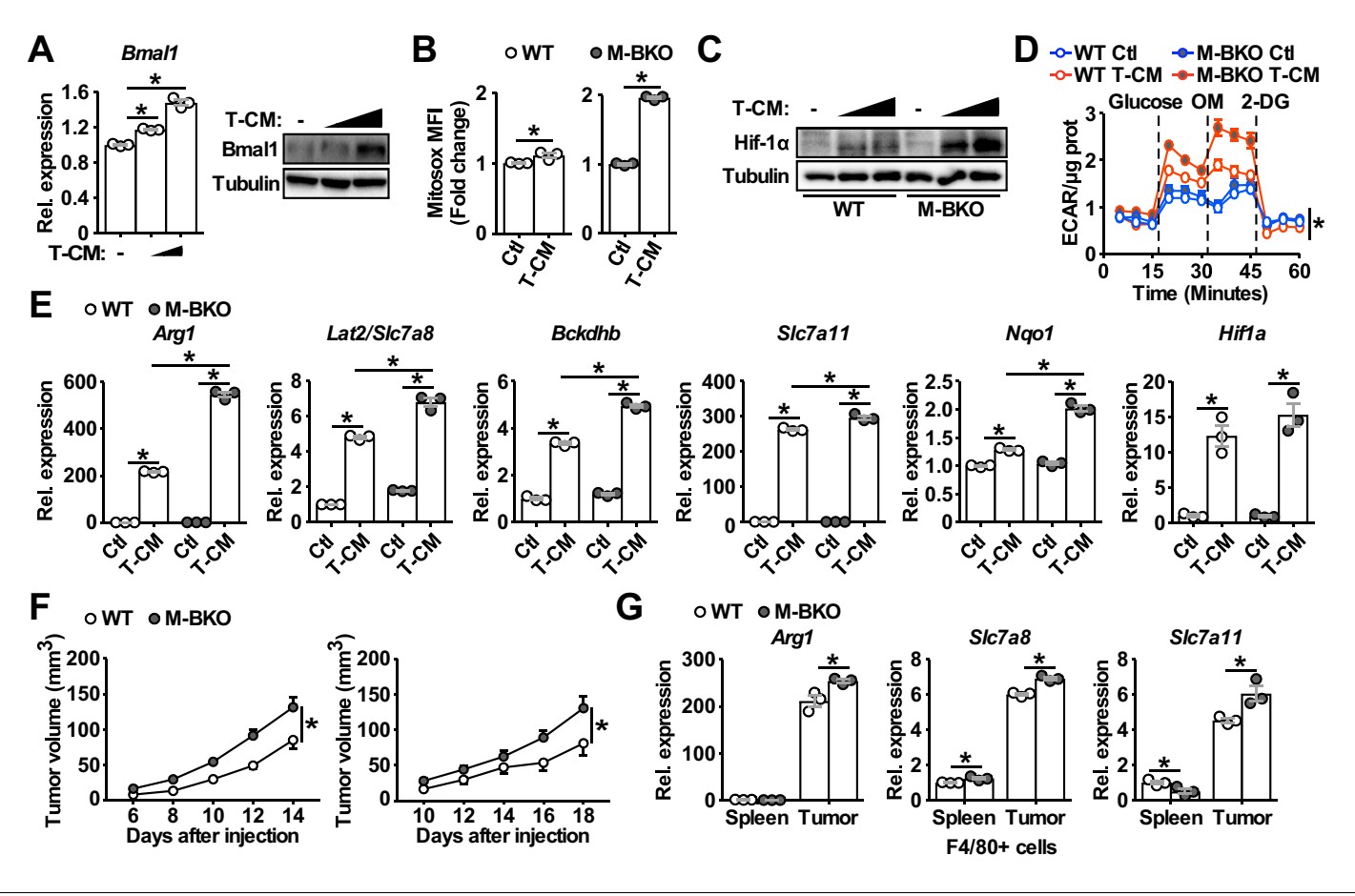

**Figure 6.** Bmal1 regulates tumor-associated macrophage polarization. (**A**) Bmal1 gene expression (left panel) and protein levels (right panel) in WT macrophages treated with control medium or increasing doses of B16-F10 tumor-conditioned medium (T-CM, diluted 1:3 or 1:1 with control medium) for 8 hr. N = 3 biological replicates for qPCR, statistical analysis was performed using Student's T test. (**B**) Measurement of mROS using MitoSox Red (mean fluorescent intensity, MFI) in mitochondria from macrophages treated with control medium or T-CM diluted 1:1 with control medium for 1 hr. N = 3 biological replicates, statistical analysis was performed using Student's T test. (**C**) Hif-1α protein levels in WT and M-BKO macrophages treated with control medium, T-CM diluted 1:1 with control medium, or undiluted T-CM for 4 hr. (**D**) Glycolytic stress test in macrophages pretreated with control medium or T-CM diluted 1:1 with control medium for 4 hr. N = 5 biological replicates. Statistical analysis was performed using two-way ANOVA comparing T-CM-treated M-BKO with WT cells across the time course. (**E**) Relative expression of genes involved in amino-acid metabolism and oxidative stress response in macrophages treated with control medium or T-CM diluted 1:3 with control medium for 8 hr, as determined by qPCR. N = 3 biological replicates, statistical analysis was performed using Student's T test. (**F**) Tumor volume in male (left) and female (right) WT and M-BKO mice. 300,000 B16-F10 cells were injected subcutaneously into the right flank. N = 18 (male) and 8 (female) mice, statistical analysis was performed using two-way ANOVA for WT vs M-BKO mice across the time course. (**G**) Gene expression for F4/80[+] cells isolated from B16-F10 tumors or spleens of female mice 14 days after injection. Tissues from six mice per genotype were pooled into three groups for leukocyte isolation. Statistical analysis was performed using Student's T test. Data are presented as mean ± S.E.M. *, p<0.05. Experiments were repeated at least twice.

The online version of this article includes the following figure supplement(s) for figure 6:

**Figure supplement 1.** Increased mROS levels and glucose uptake in M-BKO TAMs.

mitochondrial biogenesis (e.g., *Pparg*) is downregulated rapidly after M1 stimulation and rebounds after between 8–12 hr, when *Bmal1* and *Ppard* expression is induced (*Figure 1*). Several lines of evidence indicate that Bmal1 plays a key role in restoring mitochondrial function and in modulating a Hif-1α-mediated inflammatory response. The expression of transcription factors that are known to control mitochondrial bioenergetics (i.e., Pparγ and Pparδ) is downregulated by M-BKO. Macrophages that are deficient in *Bmal1* are unable to sustain mitochondrial function upon M1 stimulation

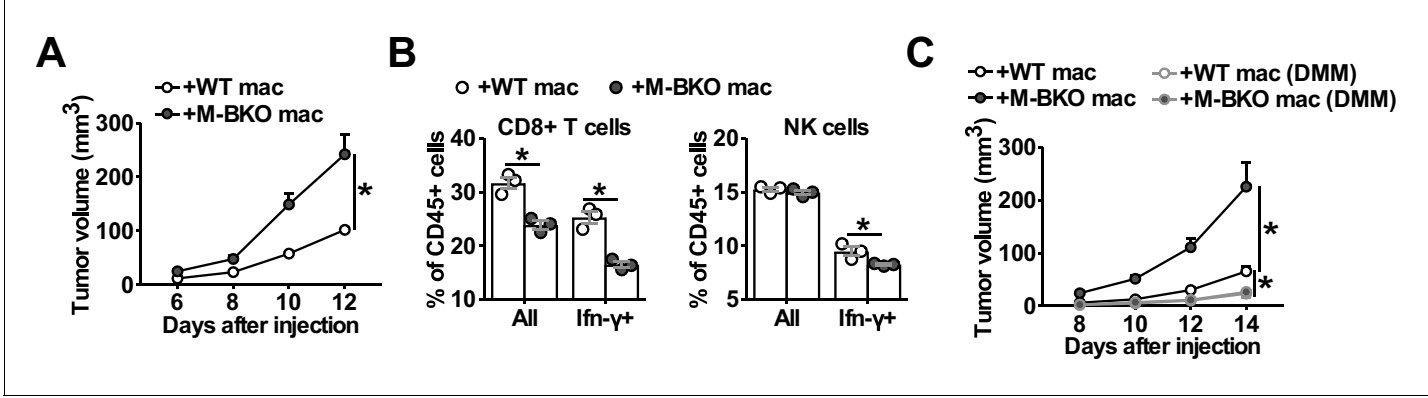

**Figure 7.** Macrophage Bmal1 modulates the antitumor activity. (A) Tumor volume in WT male mice co-injected with 500,000 B16-F10 cells and either 500,000 WT or M-BKO macrophages as indicated. N = 22 mice, statistical analysis was performed using two-way ANOVA to compare WT vs M-BKO macrophage co-injection across the time course. (B) Flow cytometric analysis of tumor-infiltrating CD8[+] T cells (CD45[+]CD3[+]CD8a[+] cells, left panel) and NK cells (CD45[+]CD3[−]NK1.1[+] cells, right panel) stimulated ex vivo with phorbol 12-myristate 13-acetate and ionomycin for Ifn-γ co-staining. Tumors represented in panel (A) were pooled into three groups prior to isolation of infiltrating leukocytes for flow cytometry. Statistical analysis was performed using Student's T test. (C) Tumor volume in WT male mice co-injected with 500,000 B16-F10 cells and either 500,000 WT or M-BKO macrophages, supplemented without or with dimethylmalonate (DMM, approximately 150 mg/kg body weight per day in mouse diet). N = 8 mice, statistical analysis was performed using two-way ANOVA to compare WT vs M-BKO macrophage co-injection on control diet, or to compare WT macrophage co-injection on the control diet vs WT or M-BKO macrophage co-injection on the DMM-supplemented diet across the time course. Data are presented as mean ± S.E.M. *, p<0.05. Experiments were repeated at least twice.

The online version of this article includes the following figure supplement(s) for figure 7:

**Figure supplement 1.** Macrophage Bmal1 regulates tumor growth in a cell-autonomous manner.

and fail to recover from suppressed mitochondrial respiration 24 hr following acute LPS treatment. By contrast, Bmal1 gain-of-function in RAW 246.7 macrophages promotes oxidative metabolism. It is interesting to note that the dysregulated mitochondrial respiration phenotype of M-BKO macrophages occurs as early as 2 hr after M1 activation, whereas Bmal1 protein accumulation peaks at 12 hr. This early phase of regulation could be mediated by downstream pathways of the circadian regulatory network. RNA-seq analyses reveal that components of the molecular clock are induced by M1 (*Figure 1—figure supplement 1A*). Notably, Nr1d1/Nr1d2 have been shown to regulate macrophage inflammatory gene expression negatively (*Lam et al., 2013*) and promote mitochondrial function in skeletal muscle (*Woldt et al., 2013*). The induction of *Nr1d2* by M1 was almost completely abolished in M-BKO macrophages (*Figure 1D*). In concert, the expression of *Cx3cr1*, a direct target suppressed by Nr1d1/Nr1d2 (*Lam et al., 2013*), is higher in M1-activated M-BKO macrophages than in WT cells (*Figure 5—source data 1*). This suppressive effect of Nr1d1/Nr1d2 may dampen the inflammatory damage to mitochondrial function at the initial stage of M1 activation. Another potential mechanism is through enhanced Hif-1α activity by M-BKO, which is evident after 4 hr of M1 activation (*Figure 4A*). Hif-1α suppresses mitochondrial respiration through multiple mechanisms (*Thomas and Ashcroft, 2019*). For instance, it regulates the glycolytic program favoring lactate production and controls the expression of Bnip3 to promote mitophagy. M-BKO macrophages exhibit a higher lactate production rate 4 hr after M1 stimulation (*Figure 3D*) and this phenotype was observed as early as 2 hr after M1 stimulation (data not shown). Bnip3 protein accumulation is also increased, although appreciable amounts of Bnip3 protein could only be detected at 8 hr after M1 treatment (*Figure 2—figure supplement 1A*). Bmal1 is best known for its role in circadian regulation and has been shown to control rhythmic monocyte recruitment, which plays a key role in the immune response against pathogens and in limiting infection-associated inflammatory damage (*Nguyen et al., 2013*). Our data suggest that LPS or M1 stimulation could 'reset the clock' by inducing/resynchronizing the expression of *Bmal1*. In this context, Bmal1 may regulate energy metabolism to support diurnal monocyte trafficking, and may control the timing of

glycolytic to oxidative metabolism transition during M1 activation that dictates the extent of Hif-1α activation and the associated inflammatory response.

Both Bmal1 and Hif-1α belong to the basic helix-loop-helix (bHLH) transcription factor family and have similar domain structures. However, they appear to regulate opposing metabolic programs, with Hif-1α serving as a master regulator of aerobic glycolysis and Bmal1 as a positive regulator of oxidative metabolism (*Figure 7—figure supplement 1E*). The crosstalk between these two bHLH transcription factors is in part mediated by succinate and SDH/complex II-facilitated mROS production. Succinate is one of the entry points for anerplerosis that attempts to replenish the TCA cycle metabolites that are depleted by disruption of mitochondrial oxidative metabolism. Increased protein or amino-acid catabolism provides a source of anerplerotic reactions. Succinate accumulation and the subsequent oxidation to fumarate, however, generate mROS, which stabilize Hif-1α protein and thus drive aerobic glycolysis. Our data suggest that amino-acid metabolism appears to be down- and upregulated by Bmal1 and Hif-1α, respectively, as demonstrated by the regulation of *Arg1* and *Slc7a8* gene expression. As described above, Hif-1α has also been shown to regulate Bnip3-mediated mitophagy that reduces mitochondrial oxidative capacity (*Zhang et al., 2008*). Therefore, Bmal1-controlled mitochondrial metabolism provides a break to this feedforward cycle that limits inflammatory damage. In line with this, previous work has demonstrated that myeloid *Bmal1* knockout mice have reduced survival rate upon *L. monocytogenes* infection (*Nguyen et al., 2013*). These observations indicate a tightly regulated metabolic program in the macrophage that executes effector functions and places Bmal1-regulated mitochondrial metabolism at the center of an orderly and balanced immune response.

Despite being characterized as M2-like, TAMs share several common features with M1-activated macrophages: both function under nutrient-restricted conditions and Hif-1α is required for their activation. Previous studies implicate a glycolytic preference for TAMs in breast, thyroid, and pancreatic cancer (*Arts et al., 2016*; *Liu et al., 2017*; *Penny et al., 2016*). Our data confirm that T-CM treatment enhances glycolysis in the macrophage accompanied by increased Hif-1α protein (*Figure 6C, D*). *Arg1*, originally defined as an M2 marker, is a *bona fide* target of Hif-1α that is upregulated in TAMs and M1 macrophages. Arg1 is involved in the urea cycle that detoxifies ammonia, and its induction supports the upregulation of amino-acid catabolism. M-BKO macrophages show increased mROS, glycolytic metabolism, Hif-1α stabilization and upregulation of *Arg1* and *Slc7a8* upon treatment with T-CM. Dysregulated amino-acid metabolism has been shown to impact immune cell activation. Arginine depletion impairs lymphocyte function, as arginine is required for effector T cell and NK cell proliferation and maintenance (*Geiger et al., 2016*; *Lamas et al., 2012*; *Steggerda et al., 2017*). Slc7a8 transports neutral amino acids, including branched-chain amino acids that are essential for lymphocyte activation and cytotoxic function (*Sinclair et al., 2013*; *Tsukishiro et al., 2000*). Therefore, the increased amino-acid utilization by M-BKO macrophages may contribute to the observed reduction in populations of Ifn-γ-producing CD8[+] T and NK cells in tumor-infiltrating CD45[+] leukocytes (*Figure 7B* and *Figure 7—figure supplement 1B*). The fact that amino-acid or protein metabolism and oxidative stress genes (*Arg1*, *Slc7a8* and *Slc7a11*) are upregulated in TAMs, compared to splenic macrophages (*Figure 6G*), supports the notion that energetic stress is also a key determinant of TAM polarization. As a proof-of-principle approach, we show that DMM treatment blocks T-CM induced Hif-1α protein stabilization in vitro and suppresses tumor growth in vivo. Thus, M1 macrophages opt for an inefficient way to produce ATP, whereas TAMs are limited in energy allocations. Both of these processes result in an energetically challenged state in which Bmal1–Hif-1α crosstalk controls the metabolic adaptation that shapes macrophage polarization. Future studies investigating mechanisms that harness this energetic stress may identify means to modulate immune cell functions effectively.

## Materials and methods

### Key resources table

| Reagent type (species) or resource | Designation | Source or reference | Identifiers | Additional information |
|---|---|---|---|---|

*Continued on next page*

Continued

| Reagent type (species) or resource | Designation | Source or reference | Identifiers | Additional information |
|---|---|---|---|---|
| Strain, strain background (*Mus musculus*) | B6.129S4(CG)-*Arntl*$^{tm1Weit}$/J | The Jackson Laboratory | JAX: 007668 | Carry loxp sites flanking exon 8 of the *Bmal1* gene |
| Strain, strain background (*Mus musculus*) | B6.129-*Hif1a*$^{tm3Rsjo}$/J | The Jackson Laboratory | JAX: 007561 | Carry loxp sites flanking exon 2 of the *Hif1a* gene |
| Strain, strain background (*Mus musculus*) | B6.129P2-*Lyz2*$^{tm1(cre)Ifo}$/J | The Jackson Laboratory | JAX: 004781 | Myeloid-specific Cre recombinase expression |
| Antibody | Mouse monoclonal anti-Bmal1 (clone B-1) | Santa Cruz Biotechnology | Cat# sc365645; RRID:AB_10841724 | WB (1:1000) |
| Antibody | Rabbit polyclonal anti-Hif-1α | Novus Biologicals | Cat# NB100-449; RRID:AB_10001045 | WB (1:1000) |
| Antibody | Rabbit monoclonal anti-Bnip3 (clone EPR4034) | Abcam | Cat# ab109362; RRID:AB_10864714 | WB (1:1000) |
| Antibody | Rabbit polyclonal anti-β-tubulin | Cell Signaling Technology | Cat# 2146; RRID:AB_2210545 | WB (1:1000) |
| Antibody | Rabbit monoclonal anti-β-Actin (clone 13E5) | Cell Signaling Technology | Cat# 4970; RRID:AB_2223172 | WB (1:1000) |
| Antibody | Rat monoclonal anti-CD45 (clone 30-F11), PerCP/Cy5.5-conjugated | Biolegend | Cat# 103132; RRID:AB_893340 | Flow cytometry (1 µL per test) |
| Antibody | Armenian hamster monoclonal anti-CD3ε (clone 145–2 C11), PE/Cy7-conjugated | Biolegend | Cat# 100320; RRID:AB_312685 | Flow cytometry (2.5 µL per test) |
| Antibody | Rat monoclonal anti-CD8a (clone 53–6.7), Alexa Fluor 700-conjugated | Biolegend | Cat# 100730; RRID:AB_493703 | Flow cytometry (0.5 µL per test) |
| Antibody | Mouse monoclonal anti-NK1.1 (clone PK136), APC-conjugated | Biolegend | Cat# 108710 | Flow cytometry (5 µL per test) |
| Antibody | Rat monoclonal anti-F4/80 (clone BM8), Alexa Fluor 488-conjugated | Biolegend | Cat# 123120; RRID:AB_893479 | Flow cytometry (5 µL per test) |
| Antibody | Rat monoclonal anti-F4/80 (clone BM8), APC-conjugated | Biolegend | Cat# 123115; RRID:AB_893493 | Flow cytometry (2.5 µL per test) |
| Antibody | Rat monoclonal anti-Ifn-γ (clone XMG1.2), PE-conjugated | ThermoFisher Scientific | Cat# 12-7311-81, RRID:AB_466192 | Flow cytometry (1 µL per test) |
| Peptide, recombinant protein | Recombinant murine interferon-γ | Peprotech | Cat# 315–05 | Used 10 ng/mL final concentration |
| Peptide, recombinant protein | Recombinant murine interleukin-4 | Peprotech | Cat# 214–14 | Used 10 ng/mL final concentration |
| Sequence-based reagent | 36b4_F | This paper | qPCR primer | AGATGCAGCAGATCCGCAT |
| Sequence-based reagent | 36b4_R | This paper | qPCR primer | GTTCTTGCCCATCAGCACC |
| Sequence-based reagent | Arg1_F | This paper | qPCR primer | CGTAGACCCTGGGGAACACTAT |
| Sequence-based reagent | Arg1_R | This paper | qPCR primer | TCCATCACCTTGCCAATCCC |

*Continued on next page*

*Continued*

| Reagent type (species) or resource | Designation | Source or reference | Identifiers | Additional information |
|---|---|---|---|---|
| Sequence-based reagent | Bckdhb_F | This paper | qPCR primer | TGGGGCTCTCTACCATTCTCA |
| Sequence-based reagent | Bckdhb_R | This paper | qPCR primer | GGGGTATTACCACCTTGATCCC |
| Sequence-based reagent | Bmal1_F | This paper | qPCR primer | AGGATCAAGAATGCAAGGGAGG |
| Sequence-based reagent | Bmal1_R | This paper | qPCR primer | TGAAACTGTTCATTTTGTCCCGA |
| Sequence-based reagent | cMyc_F | This paper | qPCR primer | CAGCGACTCTGAAGAAGAGCA |
| Sequence-based reagent | cMyc_R | This paper | qPCR primer | GACCTCTTGGCAGGGGTTTG |
| Sequence-based reagent | Cry1_F | This paper | qPCR primer | CACTGGTTCCGAAAGGGACTC |
| Sequence-based reagent | Cry1_R | This paper | qPCR primer | CTGAAGCAAAAATCGCCACCT |
| Sequence-based reagent | Gclc_F | This paper | qPCR primer | CATCCTCCAGTTCCTGCACA |
| Sequence-based reagent | Gclc_R | This paper | qPCR primer | ATGTACTCCACCTCGTCACC |
| Sequence-based reagent | Hif1a_F | This paper | qPCR primer | GAACGAGAAGAAAAATAGGATGAGT |
| Sequence-based reagent | Hif1a_R | This paper | qPCR primer | ACTCTTTGCTTCGCCGAGAT |
| Sequence-based reagent | Hmox1_F | This paper | qPCR primer | CAGAGCCGTCTCGAGCATAG |
| Sequence-based reagent | Hmox1_R | This paper | qPCR primer | CAAATCCTGGGGCATGCTGT |
| Sequence-based reagent | Il1b_F | This paper | qPCR primer | AGCTTCAGGCAGGCAGTATC |
| Sequence-based reagent | Il1b_R | This paper | qPCR primer | AAGGTCCACGGGAAAGACAC |
| Sequence-based reagent | Ldha_F | This paper | qPCR primer | GCGTCTCCCTGAAGTCTCTT |
| Sequence-based reagent | Ldha_R | This paper | qPCR primer | GCCCAGGATGTGTAACCTTT |
| Sequence-based reagent | Mgl2_F | This paper | qPCR primer | ccttgcgtttgtcaaaacatgac |
| Sequence-based reagent | Mgl2_R | This paper | qPCR primer | ctgaggcttatggaactgaggc |
| Sequence-based reagent | Mmsdh_F | This paper | qPCR primer | GAGGCCTTCAGGTGGTTGAG |
| Sequence-based reagent | Mmsdh_R | This paper | qPCR primer | GATAGATGGCATGGTCTCTCCC |
| Sequence-based reagent | Nfe2l2_F | This paper | qPCR primer | GGTTGCCCACATTCCCAAAC |
| Sequence-based reagent | Nfe2l2_R | This paper | qPCR primer | GCAAGCGACTCATGGTCATC |
| Sequence-based reagent | Nfkb1_F | This paper | qPCR primer | CCTGCTTCTGGAGGGTGATG |
| Sequence-based reagent | Nfkb1_R | This paper | qPCR primer | GCCGCTATATGCAGAGGTGT |

*Continued on next page*

*Continued*

| Reagent type (species) or resource | Designation | Source or reference | Identifiers | Additional information |
|---|---|---|---|---|
| Sequence-based reagent | Nqo1_F | This paper | qPCR primer | TCTCTGGCCGATTCAGAGTG |
| Sequence-based reagent | Nqo1_R | This paper | qPCR primer | TGCTGTAAACCAGTTGAGGTTC |
| Sequence-based reagent | Nr1d2_F | This paper | qPCR primer | TCATGAGGATGAACAGGAACCG |
| Sequence-based reagent | Nr1d2_R | This paper | qPCR primer | CGGCCAAATCGAACAGCATC |
| Sequence-based reagent | Ppard_F | This paper | qPCR primer | CAGCCTCAACATGGAATGTC |
| Sequence-based reagent | Ppard_R | This paper | qPCR primer | TCCGATCGCACTTCTCATAC |
| Sequence-based reagent | Pparg_F | This paper | qPCR primer | CAGGAGCCTGTGAGACCAAC |
| Sequence-based reagent | Pparg_R | This paper | qPCR primer | ACCGCTTCTTTCAAATCTTGTCTG |
| Sequence-based reagent | Slc7a2_F | This paper | qPCR primer | CCCGGGATGGCTTACTGTTT |
| Sequence-based reagent | Slc7a2_R | This paper | qPCR primer | AGGCCATCACAGCAGAAATGA |
| Sequence-based reagent | Slc7a8_F | This paper | qPCR primer | GAACCACCCGGGTTCTGAC |
| Sequence-based reagent | Slc7a8_R | This paper | qPCR primer | TGATGTTCCCTACAATGATACCACA |
| Sequence-based reagent | Slc7a11_F | This paper | qPCR primer | ATCTCCCCCAAGGGCATACT |
| Sequence-based reagent | Slc7a11_R | This paper | qPCR primer | GCATAGGACAGGGCTCCAAA |
| Sequence-based reagent | Stat3_F | This paper | qPCR primer | TGGCAGTTCTCGTCCAC |
| Sequence-based reagent | Stat3_R | This paper | qPCR primer | CCAGCCATGTTTTCTTTGC |
| Sequence-based reagent | Bmal1_F | This paper | Cloning primers | GGCGAATTCGCGGA CCAGAGAATGGAC |
| Sequence-based reagent | Bmal1_R | This paper | Cloning primers | GGGCTCGAGCTACA GCGGCCATGGCAA |
| Cell line (*Mus musculus*) | RAW264.7 macrophages | ATCC | qPCR primer | TIB-71 |
| Cell line (*Mus musculus*) | B16-F10 melanoma cells | ATCC | qPCR primer | CRL-6475 |
| Recombinant DNA reagent | pBABE-puro retroviral expression vector (plasmid) | Addgene | Plasmid #1764 | |
| Recombinant DNA reagent | pBABE-Bmal1 (plasmid) | This study | | For stable overexpression of mouse *Bmal1* |
| Chemical compound, drug | Lipopolysaccharides from *Escherichia coli* K-235 | Sigma-Aldrich | Cat# L2143 | Final concentration used varied as indicated |
| Chemical compound, drug | Dimethyl malonate | Sigma-Aldrich | Cat# 136441; CAS: 108-59-8 | Used 10 mM final concentration |
| Chemical compound, drug | Sodium succinate dibasic | Sigma-Aldrich | Cat# 14160; CAS: 150-90-3 | Used 10 mM final concentration |
| Chemical compound, drug | N-acetyl-L-cysteine | Sigma-Aldrich | Cat# A9165; CAS: 616-91-1 | Final concentration used varied as indicated. |

*Continued on next page*

*Continued*

| Reagent type (species) or resource | Designation | Source or reference | Identifiers | Additional information |
|---|---|---|---|---|
| Chemical compound, drug | 2-deoxy-D-glucose | Sigma-Aldrich | Cat# D8375; CAS: 154-17-6 | Used 50 mM final concentration |
| Chemical compound, drug | Quant-iT ribogreen RNA reagent | ThermoFisher Scientific | Cat# R11491 | |
| Chemical compound, drug | Mitotracker green FM | ThermoFisher Scientific | Cat# M7514 CAS: 201860-17-5 | Used 100 nM final concentration |
| Chemical compound, drug | Mitosox red superoxide indicator | ThermoFisher Scientific | Cat# M36008 | Used 5 µM final concentration |
| Chemical compound, drug | 2-NBDG | ThermoFisher Scientific | Cat# N13195 | Used 10 µM final concentration |
| Chemical compound, drug | Fixable viability dye eFluor 455uv | ThermoFisher Scientific | Cat# 65-868-14 | Flow cytometry (0.5 µL per test) |
| Chemical compound, drug | Fixable viability dye eFluor 506 | ThermoFisher Scientific | Cat# 65-0866-14 | Flow cytometry (0.5 µL per test) |
| Chemical compound, drug | Oligomycin | Abcam | Cat# ab141829; CAS: 1404-19-9 | Used 2 µM final concentration |
| Chemical compound, drug | Rotenone | Abcam | Cat# ab143145 CAS: 83-79-4 | Used 1 µM final concentration |
| Chemical compound, drug | Antimycin A | Sigma-Aldrich | Cat# A8674; CAS: 1397-94-0 | Used 1 µM final concentration |
| Chemical compound, drug | Carbonyl cyanide-4-(trifluoromethoxy) phenylhydrazone (FCCP) | Santa Cruz Biotechnology | Cat# sc203578 CAS: 370-86-5 | Used 0.5 µM final concentration |
| Chemical compound, drug | Phorbol 12-myristate 13-acetate (PMA) | Sigma-Aldrich | Cat# P8139; CAS: 16561-29-8 | Used 20 ng/mL final concentration |
| Chemical compound, drug | Ionomycin calcium salt from *Streptomyces conglobatus* | Sigma-Aldrich | Cat# I0634; CAS: 56092-82-1 | Used 1 µg/mL final concentration |
| Chemical compound, drug | Brefeldin A | Cell Signaling Technology | Cat# 9972S; CAS: 20350-15-6 | Used 10 µg/mL final concentration |
| Chemical compound, drug | BCA protein assay kit | ThermoFisher Scientific | Cat# 23227 | |
| Chemical compound, drug | Seahorse XF24 FluxPak | Agilent | Cat# 102070–00 | |
| Chemical compound, drug | Foxp3/Transcription factor staining buffer set | ThermoFisher Scientific | Cat# 00-5523-00 | |
| Chemical compound, drug | TruSeq stranded mRNA library prep kit | Illumina | Cat# RS-122–2101 | |
| Chemical compound, drug | Dynabeads sheep anti-rat IgG | ThermoFisher Scientific | Cat # 11035 | |
| Software, algorithm | STRING | *von Mering, 2004* | https://string-db.org/ | |
| Software, algorithm | DAVID | *Huang et al., 2009* | https://david-d.ncifcrf.gov/ | |
| Other | Thioglycollate medium | Sigma-Aldrich | Cat# T9032 | |
| Other | Bovine serum albumin, fatty acid free | Gemini Bio-Products | Cat# 700–107P | |
| Other | Collagen I protein, rat tail | ThermoFisher Scientific | Cat# A1048301 | |
| Other | Collagenase type IV | ThermoFisher Scientific | Cat# 17104019 | |
| Other | Recombinant DNase I, RNase-free | Affymetrix | Cat# 78411 | |

## Reagents

Lipopolysaccharide (LPS) from *Escherichia coli* strain K-235 (L2143) was from Sigma-Aldrich. Recombinant murine Ifn-γ (315-05) and Il-4 (214–14) were from Peprotech. The ETC complex II inhibitor dimethyl malonate (136441) and the L-type amino acid transport inhibitor 2-amino-2-norbornanecarboxylic acid, or BCH, (A7902) were from Sigma-Aldrich.

## Animals

All animal studies were approved by the Harvard Medical Area Standing Committee on Animal Research. Animals were housed in a pathogen-free barrier facility at the Harvard T.H. Chan School of Public Health. *Bmal1*$^{fl/fl}$ (stock # 007668), *Hif1a* $^{fl/fl}$ (stock # 007561), and *Lyz2-Cre* (stock # 004781) mice in the C57BL/6J background were obtained from Jackson laboratories and were originally contributed by Drs Charles Weitz, Dmitriy Lukashev, and Irmgard Foerster, respectively. Floxed mice were crossed with *Lyz2-Cre* mice to generate myeloid-specific *Bmal1* and *Hif1a* knockout mice. Myeloid-specific *Bmal1* knockout mice were crossed with *Hif1a*$^{fl/fl}$ mice, and the resulting heterozygotes were crossed to generate myeloid-specific *Bmal1* and *Hif1a* double-knockout mice. The genotypes were validated by both DNA genotyping and mRNA expression. Gender- and age-matched mice of between 8–24 weeks of age were used for experiments. Similar results were obtained from male and female mice.

## Bone marrow-derived macrophage (BMDM) differentiation and cell culture

Macrophages were differentiated from primary mouse bone marrow from the femur and tibia using differentiation medium containing 30% L929-conditioned medium, 10% FBS, and pen-strep solution in low-glucose DMEM in 15-cm Petri dishes. Media were changed every three days, and cells were lifted, counted, and plated in final format in tissue culture plates on days 7–8 of differentiation. For experiments, primary macrophages were maintained in low glucose DMEM containing 10% FBS and pen-strep. For M1 activation, macrophages were primed with 10 ng/mL Ifn-γ for 10–12 hr and subsequently stimulated with 10 ng/mL of *E. coli* LPS at the start of each experiment. Macrophages with Ifn-γ priming but without LPS were used as the control for M1 activation.

## Peritoneal and splenic macrophage isolation and culture

For peritoneal macrophage isolation, mice aged 2–4 months were *i.p.* injected with 3 mL of 3% thioglycollate (Sigma-Aldrich, T9032). After 3 days, mice were euthanized, and peritoneal cells were recovered by lavage. For isolation of splenic macrophages, mice were euthanized and spleens were dissected and mashed in growth medium (high glucose DMEM with 10% FBS) and passed through a 70 µm strainer. Cells were pelleted and resuspended in red blood cell lysis buffer. Monocytes and lymphocytes were recovered using Ficoll-Paque Plus density gradient medium (GE Healthcare Life Sciences, 17144002) according to the manufacturer's instructions, and suspension cells (lymphocytes) were washed away prior to experiments.

## LPS synchronization of Bmal1 expression

To synchronize *Bmal1* gene and protein expression with LPS (or LPS shock), BMDMs or mouse embryonic fibroblasts (MEFs) were given fresh culture medium with 2% FBS and 100 ng/mL LPS for 1 hr and then given fresh medium with 2% FBS without LPS. *Bmal1* expression was tracked following LPS removal. For M1 or LPS induction of *Bmal1* expression, cells were primed with or without 10 ng/mL Ifn-γ for 10–12 hr in DMEM, 10% FBS and subsequently stimulated with 10 ng/mL of LPS without changing the medium (time zero).

## Cell lines

MEFs were isolated from WT C57/BL6J mouse embryos and immortalized using the 3T3 protocol as previously described (*Xu, 2005*). For experiments, immortalized MEFs were maintained in growth medium containing high glucose DMEM, 10% FBS. RAW264.7 mouse macrophages (TIB-71) and B16-F10 mouse melanoma cells (CRL-6475) were purchased from ATCC and experiments were conducted using early passages. Mycoplasma contamination was monitored using PCR-based methods. For generation of stable Bmal1-overexpressing RAW264.7 cells, the *Bmal1*

coding sequence was cloned from mouse embryonic cDNA (forward primer: 5′ GGCGAA TTCGCGGACCAGAGAATGGAC 3′; reverse primer: 5′ GGGCTCGAGCTACAGCGGCCATGGCAA 3′) and subcloned into the pBABE retroviral expression vector (Addgene, 1764). Retroviral vectors were transfected into Phoenix packaging cells, followed by collection of supernatants containing retroviruses. RAW264.7 macrophages were incubated with retroviral supernatants with 4 μg/mL polybrene, and infected cells were selected with 4 μg/mL puromycin. Control cells were transduced with the empty pBABE vector.

## Syngeneic tumor model and tumor measurement

Male and female WT and M-BKO mice aged 10–12 weeks were subcutaneously injected in the right flank with 300,000 B16-F10 mouse melanoma cells. For co-injection experiments, 500,000 B16-F10 cells were mixed with either 500,000 WT or M-BKO BMDMs (differentiation for 6 day) in the right and left flanks, respectively. Tumor dimensions were measured every two days by caliper after all mice had palpable tumors, and tumor volume was calculated as LxWxWx0.52 as previously described (*Colegio et al., 2014*). For DMM treatment, mice were switched to a soft pellet, high fat diet (Bio-Serv, F3282) so that DMM could be mixed with the diet using a blender. The tumor growth rate was slower on high fat diet (*Figure 7C*) compared to normal chow (*Figure 7A*).

## RNA sequencing

RNA-seq was performed on RNA from three biological replicates per treatment. Sequencing and raw data processing were conducted at the Institute of Molecular Biology (IMB) Genomics Core and IMB Bioinformatics Service Core, respectively, at Academia Sinica (Taipei, Taiwan, ROC). In brief, RNA was quantified using the Quant-iT ribogreen RNA reagent (ThermoFisher, R11491), and RNA quality was determined using a Bioanalyzer 2100 (Agilent; RIN > 8, OD 260/280 and OD 260/ 230 > 1.8). RNA libraries were prepared using the TruSeq Stranded mRNA Library Preparation Kit (Illumina, RS-122–2101). Sequencing was analyzed with an Illumina NextSeq 500 instrument. Raw data were analyzed using the CLC Genomics Workbench. Raw sequencing reads were trimmed by removing adapter sequences, low-quality sequences (Phred quality score of <20) and sequences >25 bp in length. The trimmed reads were then mapped to the mouse genome assembly (mm10) from University of California, Santa Cruz, using the following parameters: mismatches = 2, minimum fraction length = 0.9, minimum fraction similarity = 0.9, and maximum hits per read = 5. Gene expression was determined by the number of transcripts per kilobase million. Functional annotation clustering of differentially regulated genes was done using DAVID (https://david-d.ncifcrf.gov/), and the interaction maps of transcriptional regulators that were induced or repressed by M1 activation that are shown in *Figure 1—figure supplement 1A* were generated using STRING (https://string-db.org/). Significantly changed genes were determined by p<0.05 and FDR <0.05. Data have been deposited in GEO under the accession number GSE148510.

## qPCR

Relative gene expression was determined by real-time qPCR with SYBR Green. The expression of the ribosomal subunit *36b4* (*Rplp0*) was used as an internal control to normalize expression data. Primer information is described in the 'Key resources table'.

## Western blot

Standard Tris-Glycine SDS-PAGEs were run and transferred to PVDF membranes by wet transfer. Membranes were incubated with primary antibodies in TBST buffer with 1% BSA overnight. ECL signal was imaged using a BioRad ChemiDoc XRS+ imaging system. The antibody for Bmal1 (sc365645) was from Santa Cruz. The antibody for Hif-1α (NB100-449) was from Novus Biologicals. The antibodies for β-tubulin (2146) and β-actin (4970) were from Cell Signaling Technology.

## Extracellular flux analyses

Extracellular flux experiments were done using a Seahorse XF24 analyzer (Agilent) and FluxPaks (Agilent, 100850–001). 200,000 BMDMs, splenic/peritoneal macrophages or RAW264.7 cells were seeded into Seahorse XF24 plates for extracellular flux experiments. Minimal DMEM (pH 7.4) without phenol red and containing energy substrates as indicated was used as the assay

medium. 2% dialyzed FBS was added to media for experiments in which LPS was injected during the assay to enhance responsiveness to LPS. Assay measurements were normalized to total protein content.

## Glucose uptake assay

BMDMs were plated at a density of 1 million cells per well in 12-well plates and stimulated as indicated. Cells were then washed with Krebs-Ringer bicarbonate HEPES (KRBH) buffer and then given 400 µL with KRBH buffer loaded with 0.8 µCi/well [$^3$H]−2-deoxyglucose (PerkinElmer, NET549A001MC) and 0.5 mM unlabeled 2-deoxyglucose and incubated at 37˚C for 30 min. 10 µL of 1.5 mM Cytochalasin B (Cayman Chemical, 11328) was then added to stop glucose uptake. 400 µL of lysate was used to measure levels of [$^3$H]−2-deoxyglucose by a scintillation counter, and the remaining lysate was used to measure total protein content for normalization.

## Measurement of lactic acid secretion

Lactic acid was measured in the supernatants of BMDMs using the Biovision Lactate Colorimetric Kit (K627) according to the manufacturer's protocol. Readings were normalized to total cellular protein content.

## Mitochondrial isolation

Mitochondria were isolated from primary BMDMs by differential centrifugation. In brief, cells were resuspended in 500 µL of ice-cold mitochondrial isolation buffer consisting of 70 mM sucrose, 50 mM Tris, 50 mM KCl, 10 mM EDTA, and 0.2% fatty-acid free BSA (pH 7.2) and then extruded through 29-gauge syringes 20 times. Lysates were spun at 800 g to pellet nuclei, and supernatants were spun at 8000 g to isolate mitochondria. Pelleted mitochondria were washed once more with 500 µL of mitochondria isolation buffer. Total mitochondrial protein content was determined by BCA assay.

## ETC activity assays in isolated mitochondria

The activities of ETC complexes I–IV were measured in isolated mitochondria using colorimetric assays as previously described (*Spinazzi et al., 2012*) with modifications. In brief, 15 µg of mitochondria were loaded per reaction for complexes III and IV, and 30 and 50 µg were used for complexes II and I, respectively. Complex I activity was determined by the decrease in absorbance at 340 nm corresponding to reduction of ubiquinone by electrons from NADH. Complex II activity was determined by the decrease in absorbance at 600 nm corresponding to reduction of decylubiquinone by electrons from succinate. Complex III activity was determined by the increase in absorbance at 550 nm corresponding to reduction of cytochrome C. Complex IV activity was determined by decrease in absorbance at 550 nm corresponding to oxidation of cytochrome C.

## Flow cytometry

For flow cytometry, BMDMs were seeded into low-attachment plates for the indicated treatments and resuspended by pipetting. Mitochondrial content in BMDMs was determined by flow cytometry of live cells stained with 100 µM Mitotracker Green FM (ThermoFisher, M7514) according to the manufacturer's instructions.

For flow cytometry of tumor-infiltrating lymphocytes, cells were stimulated ex vivo with 20 ng/mL phorbol 12-myristate 13-acetate (PM), (Sigma-Aldrich, P8139) and 1 µg/mL ionomycin (Sigma-Aldrich, I0634) for 4 hr and co-treated with brefeldin A (Cell Signaling Technology, 9972) to inhibit cytokine release. Cells were stained with the fixable viability dye (eFluor 455uv [ThermoFisher, 65-868-14] or eFluor 506 [ThermoFisher, 65-0866-14]) for 20 min at 4˚C in PBS, washed, and incubated with antibodies against the indicated surface antigens for 30 min at 4˚C in fluorescence-activated cell sorting (FACS) buffer (2% FBS and 1 mM EDTA in PBS). Cells were then washed twice and fixed with 2% paraformaldehyde for 1 hr at 4˚C, and resuspended and stored in FACS buffer prior to downstream analysis. Immediately before flow cytometric analysis, cells were permeabilized for intracellular staining using the Foxp3/Transcription factor staining buffer set (ThermoFisher, 00-5523-00) according to the manufacturer's instructions. Of viable cells, CD8$^+$ T cells were identified as CD45$^+$-CD3$^+$CD8a$^+$ cells and NK cells were identified by CD45$^+$ CD3$^-$ NK1.1$^+$ staining. To determine

glucose uptake and mROS production by primary TAMs, live cells were stained with either 5 µM Mitosox Red (ThermoFisher, M36008) or 10 µM 2-NBDG (ThermoFisher, N13195) for 20 min in RPMI 1640 medium at 37°C, then washed and resuspended in FACS buffer for flow cytometry. F480$^+$ cells were gated from live CD45$^+$ cells to measure the mean fluorescent intensities of Mitosox or 2-NBDG. Antibodies for PerCp/Cy5.5-conjugated CD45 (103132), PE/Cy7-conjugated CD3e (100320), Alexa Fluor 700-conjugated CD8a (100730), APC-conjugated NK1.1 (108710) Alexa 488-conjugated F480 (123120) and APC-conjugated F480 (123115) were from Biolegend. The antibody for PE-conjugated Ifn-γ (12-7311-81) was from ThermoFisher Scientific.

To measure ROS production by isolated mitochondria, 15 µg of mitochondria were resuspended in 500 µL mitochondrial isolation buffer containing 5 µM MitoSox Red and 100 µM MitoTracker Green FM with or without 10 mM sodium succinate. Mitochondria were incubated for 20 min at room temperature, washed with isolation buffer, and resuspended for flow cytometry. Mitochondria were identified by side scatter and positive MitoTracker Green staining for measurement of mean MitoSox Red intensity per population.

## Steady-state metabolomics

Untargeted metabolomics analysis using GC-TOF mass spectrometry was conducted by the West Coast Metabolomics Center at UC Davis. In brief, 10 million cells were lifted, pelleted, and washed twice with PBS for each replicate. Cell lysates were homogenized by metal bead beating, and metabolites were extracted using 80% methanol. Following extraction, cell pellets were solubilized using Tris-HCl urea buffer (pH 8.0) containing 1% SDS to measure cellular protein content for each sample. All metabolite readings were normalized to total protein content.

## Collection of tumor-conditioned medium

Mice bearing subcutaneous B16-F10 tumors were sacrificed 20 days after injection with 500,000 cells. Tumors were dissected and weighed. Tumors were minced in growth medium containing 10% dialyzed FBS in high-glucose DMEM (5 mL per gram of tissue) and incubated at 37°C for 2 hr. Conditioned medium was collected and filtered through a 100 µm strainer followed by three spins at 1,000 rpm to pellet and remove residual cells and debris from the medium.

## Isolation of tumor-infiltrating immune cells

Subcutaneous mouse tumors were dissected, weighed, and then placed in six-well plates with growth medium (RPMI, 5% FBS) and minced. Minced tissues were combined into three groups per genotype, spun down in 50 mL conical tubes, and resuspended in 20 mL digestion buffer (0.5 mg/ mL collagenase IV, 0.1 mg/mL DNase I in HBSS medium). Tumors were digested at 37°C with gentle shaking for 30 min and vortexed every 10 min. Contents were filtered through a 100 mm mesh, and cells were pelleted and resuspended in 45% percoll in 1X HBSS and 1X PBS. Cells were spun at 2000 rpm at 4°C with a swing bucket rotor for 20 min. The supernatant was aspirated, and the pellet was briefly resuspended in 5 mL ACK buffer to lyse red blood cells. Last, cells were pelleted and resuspended in growth medium for downstream applications.

To isolate F4/80$^+$ cells from tumors, tissues were homogenized and processed as above to collect tumor-infiltrating leukocytes. F4/80$^+$ cells were then isolated by positive selection using a rat anti-F4/80 antibody (Biolegend, 123120) and sheep anti-rat Dynabeads (ThermoFisher Scientific, 11035) according to the manufacturer's instructions.

## Statistical analysis

All data are presented as mean ± SEM. GraphPad Prism 7 was used for statistical analyses. Two-tailed Student's t test was used for comparisons of two parameters. Two-way ANOVA was used for multi-parameter analyses for time course comparisons. Cell-based experiments were performed with 3–5 biological replicates (cell culture replicates). For tumor volume, outliers were determined using a Rout test ($p < 0.05$), and outliers were omitted from downstream experiments.

## Acknowledgements

We thank Drs D Cohen, D Sinclair, and P Weller for critical comments, A Yesian, L Dai, P Basak, R Chen, P Hu, and F Onal for technical help and the Academia Sinica (Taipei, Taiwan, ROC) for RNA sequencing/data analysis (IMB Genomics Core and Bioinformatics Service Core) and grant support (AS-106-TP-L08 [to NSL], AS-106-TP-L08-1 and AS-106-TP-L08-3). YHL was supported by funds from the Ministry of Science and Technology, Taiwan, ROC. This work was supported by grants from NIH (F31GM117854 to RKA; F31DK107256 to NHK; R01DK113791 and R21AI131659 to CHL) and the American Heart Association (16GRNT31460005 to CHL).

## Additional information

### Funding

| Funder | Grant reference number | Author |
|---|---|---|
| National Institute of Allergy and Infectious Diseases | R21AI131659 | Chih-Hao Lee |
| National Institute of General Medical Sciences | F31GM117854 | Ryan K Alexander |
| National Institute of Diabetes and Digestive and Kidney Diseases | F31DK107256 | Nelson H Knudsen |
| Ministry of Science and Technology of Taiwan | | Yae-Huei Liou |
| Academia Sinica | AS-106-TP-L08 | Nan-Shih Liao |
| American Heart Association | 16GRNT31460005 | Chih-Hao Lee |
| National Institute of Diabetes and Digestive and Kidney Diseases | R01DK113791 | Chih-Hao Lee |
| National Institute of Diabetes and Digestive and Kidney Diseases | R01DK064750 | Chih-Hao Lee |

The funders had no role in study design, data collection and interpretation, or the decision to submit the work for publication.

### Author contributions

Ryan K Alexander, Conceptualization, Data curation, Formal analysis, Investigation, Methodology, Writing - original draft, Writing - review and editing; Yae-Huei Liou, Formal analysis, Investigation, Methodology; Nelson H Knudsen, Data curation, Formal analysis, Investigation, Methodology; Kyle A Starost, Chuanrui Xu, Validation, Investigation; Alexander L Hyde, Sihao Liu, David Jacobi, Resources; Nan-Shih Liao, Formal analysis; Chih-Hao Lee, Conceptualization, Supervision, Funding acquisition, Writing - original draft, Writing - review and editing

### Author ORCIDs

Ryan K Alexander (iD) https://orcid.org/0000-0002-9204-9586
Chuanrui Xu (iD) http://orcid.org/0000-0003-3225-4083
Chih-Hao Lee (iD) https://orcid.org/0000-0002-6090-0786

### Ethics

Animal experimentation: All animal studies were approved by the Harvard Medical Area Standing Committee on Animal Research. IACUC protocol #IS00001011.

### Decision letter and Author response

Decision letter https://doi.org/10.7554/eLife.54090.sa1
Author response https://doi.org/10.7554/eLife.54090.sa2

## Additional files

### Supplementary files
• Transparent reporting form

### Data availability

Raw RNA-seq data on GEO: accession number GSE148510. All analyzed RNA-seq and metabolite data are included in the manuscript and source data for Figure 3 and Figure. 5.

The following dataset was generated:

| Author(s) | Year | Dataset title | Dataset URL | Database and Identifier |
|---|---|---|---|---|
| Alexander RK, Lee C | 2020 | RNA-sequencing of Wild-type and Bmal1 KO M1 macrophages | https://www.ncbi.nlm.nih.gov/geo/query/acc.cgi?acc=GSE148510 | NCBI Gene Expression Omnibus, GSE148510 |

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
