## [Decision Letter]

**Acceptance summary:**

The study reveals the importance of the function of the circadian clock, through BMAL1, for the appropriate activation of the inflammatory response in macrophages. As such, it advances our understanding on the coordination between mitochondrial metabolism and inflammatory cascade towards the most appropriate macrophage response.

**Decision letter after peer review:**

Thank you for submitting your article "Bmal1 integrates mitochondrial metabolism and macrophage activation" for consideration by *eLife*. Your article has been reviewed by two peer reviewers, and the evaluation has been overseen by a Reviewing Editor and Satyajit Rath as the Senior Editor. The reviewers have opted to remain anonymous.

The reviewers have discussed the reviews with one another and the Reviewing Editor has drafted this decision to help you prepare a revised submission.

Summary:

The manuscript reports a cell-autonomous role for the circadian clock gene *Bmal* in the regulation of macrophage effector functions. Specifically, the authors find that macrophages deficient in Bmal1 upregulate the glycolytic capacity at the expense of mitochondrial function upon M1-like stimulation in a Hif-1α dependent manner. They also extend their findings in an in vivo tumor model, wherein Bmal deficiency results in improved anti-tumor response. Overall, the manuscript was found to be well written, the experiments properly controlled for and the results described supported by the authors' conclusions.

Essential revisions:

1) The manuscript mostly focuses on the role of Bmal in the regulation of M1-like stimulation. As such, it remains unclear whether the findings apply specifically to the role of Bmal in regulating metabolic reprogramming solely upon macrophage activation or also to the modulation of mitochondria metabolism in basal conditions.

2) Related to the point raised above, there appears to be a disconnect between the kinetics of expression of different components upon M1 activation and the effects on mitochondria. For example, the initial identification of Bmal as a regulator of multiple transcriptional modules activated/repressed upon M1 activation is based on transcriptional profiling at a specific time point (8h after induction). Follow up studies reveal complex activation/repression kinetics and dependency on Bmal, however it is unclear how the different events relate to Bmal protein induction (which is shown to be peaking at 12h). In Figure 2 it is shown that mitochondrial content decreases as rapidly as 2h upon M1 induction when it is unknown whether Bmal induction/stabilization has already occurred. Moreover, the decrease is explained by an increase in mitophagy even though the increase in Bnip3 doesn't occur until 8h after activation. Downregulation of mitochondria biogenesis regulators PPARg and PPARd could contribute to the phenotype, but loss of Bmal seems to affect only their reactivation not the initial repression. Thus, it is unclear whether the authors interpret these results as Bmal induction in M1 activated macrophages being required for the long-term recovery of mitochondrial homeostasis or for the early decrease in mitochondrial function/content, in which case the mechanism is unclear.

3) The findings on tumor associated macrophages (TAMs) need to be extended. Specifically, do TAMs derived from tumors display a similar regulation of Bmal1-mitochondrial capacity and glucose utilization as bone marrow-derived macrophages exposed to tumor derived conditioned medium?

4) The data presented in Figure 1—figure supplement 1 indicate that the clock program downstream of Bmal is activated upon M1 induction. This raises a number of questions about the relationship between Bmal role in metabolic regulation and its well-known activity as a molecular clock. Discussing these findings in light of existing literature would be helpful for the audience to understand their possible implications in terms of circadian regulation of immunometabolism.

---

## [Author Response]

Essential revisions:1) The manuscript mostly focuses on the role of Bmal in the regulation of M1-like stimulation. As such, it remains unclear whether the findings apply specifically to the role of Bmal in regulating metabolic reprogramming solely upon macrophage activation or also to the modulation of mitochondria metabolism in basal conditions.

As the reviewers might be aware of, macrophages exhibit a spectrum of activation states. For our basal condition, cells were withdrawn from the differentiation medium (with L929 conditioned medium) and changed to DMEM, 10% FBS for the subsequence experiments. In the original (and the revised) Figure 2E, we showed the ECAR was the same between WT and M-BKO macrophages at the basal/unstimulated state.

To specifically address this question, we have performed Mitostress tests and measured mitochondrial respiration (oxygen consumption rate, OCR) under the basal condition or with a two-hour serum shock (50% FBS) that is known to synchronize circadian clock/Bmal1 expression in cultured cells. As shown in revised Figure 2—figure supplement 1C, there was no significant difference in the OCR between WT and M-BKO macrophages, with or without serum shock. We could not rule out the possibility that in vivo, Bmal1 deficiency may affect macrophage mitochondrial metabolism in an unstimulated state when energy substrate availability changes between feeding and fasting states. It would appear that under the experimental condition, Bmal1- mediated mitochondrial metabolism is more important in M1-activated macrophages.

2) Related to the point raised above, there appears to be a disconnect between the kinetics of expression of different components upon M1 activation and the effects on mitochondria. For example, the initial identification of Bmal as a regulator of multiple transcriptional modules activated/repressed upon M1 activation is based on transcriptional profiling at a specific time point (8h after induction). Follow up studies reveal complex activation/repression kinetics and dependency on Bmal, however it is unclear how the different events relate to Bmal protein induction (which is shown to be peaking at 12h). In Figure 2 it is shown that mitochondrial content decreases as rapidly as 2h upon M1 induction when it is unknown whether Bmal induction/stabilization has already occurred. Moreover, the decrease is explained by an increase in mitophagy even though the increase in Bnip3 doesn't occur until 8h after activation.

We agree there is a discordance between the peak protein accumulation and functional outcomes, which might be partly due to the limitation in detection sensitivity of immunoblotting. For instance, Hif-1α protein accumulation was detectable at 4 hour and peaked at 8 hours following M1 activation (Figure 4A). However, Hif-1α dependent glucose uptake and lactate production were evident at 4 hours (Figure 3C and 3D and Figure 4—figure supplement 1B and 1C). Bmal1 protein has already accumulated at 4 hours (Figure 1A) and a higher Hif-1α protein level was observed in M-BKO macrophages at this time point (Figure 4A). We have included in Author response image 1 the lactate production result with the 2-hour time point for the review purpose. Lactate release was induced by M1 activation at 2 hours in WT macrophages, which was further up regulated in M-BKO macrophages. As such Bmal1 is likely to have functional importance at early time points after M1 activation.

We have conducted new Mitostress test to examine mitochondrial respiration in unstimulated, 2-hour M1 or 6-hour M1 activation. As shown in the revised Figure 2—figure supplement 1B, 2h M1 activation suppressed mainly the basal OCR, while 6h M1 affected both basal and maximum respiration in WT macrophages. The suppressed mitochondrial respiration was to a greater extent in M-BKO macrophages at both time points. The comparison and statistical analysis of the basal OCR is presented in the revised Figure 2B. While the 6hr effect could be partly explained by Bnip3-mediated mitophagy, the 2hr effect could not, as pointed out by the reviewers. In fact, Figure 2C showed that given the same amount of protein, mitochondria isolated by M1 activated M-BKO had lower ETC complex activities, suggesting that M-BKO impacted mitochondrial quality as well. We have moved the Bnip3 immunoblotting to Figure 2—figure supplement 1A and modified the description by stating that the reduced mitochondrial content was accompanied by elevated Bnip3 protein levels. Potential underlying mechanisms for the early phase effect is described below.

Downregulation of mitochondria biogenesis regulators PPARg and PPARd could contribute to the phenotype, but loss of Bmal seems to affect only their reactivation not the initial repression. Thus, it is unclear whether the authors interpret these results as Bmal induction in M1 activated macrophages being required for the long-term recovery of mitochondrial homeostasis or for the early decrease in mitochondrial function/content, in which case the mechanism is unclear.

As discussed in point 2.1, results in the revised Figure 2—figure supplement 1B (and Figure 2B) confirmed that M-BKO indeed affected mitochondrial respiration in the early phase of M1 activation. To assess the recovery phase, we conducted parallel Mitostress tests with either serum shock or “LPS shock” synchronization. The inclusion of serum shock served two purposes: the first was to examine whether Bmal1 gene deletion affected basal respiration, as discussed in point 1, and the second was to provide as a reference for potential effects from medium change at the recovery phase. For these experiments, macrophages were given a 2-h serum shock (DMEM, 50% FBS) or acute LPS treatment (DMEM, 2% FBS, which induced Bmal1 gene/protein expression shown in Figure 1C), washed and cultured in DEME, 10% FBS or DMEM, 2% FBS, respectively, for additional 24 hours before the Mitostress tests. The use of 2% FBS in the acute LPS treatment was to further reduce any potential serum effects on synchronization. As mentioned in point 1, there was no genotypic difference in OCR under serum shock synchronization (revised Figure 2—figure supplement 1C). By contrast, the basal OCR of M-BKO macrophages remained suppressed 24 hours following the acute LPS treatment, while the basal OCR was completely recovered in WT macrophages (revised Figure 2—figure supplement 1D). Collectively, these results suggest that Bmal1 gene deletion impacts macrophage mitochondrial respiration both during and at the recovery phase of M1 stimulation.

For the early stage effect, we think one of the plausible mechanisms is through modulation of Hif-1α activation, as discussed in point 2.1. Hif-1α-mediated glycolysis drives pyruvate away from TCA cycle and towards lactate production. Increased lactate production was observed in M-BKO macrophages at 2hr following M1 activation (see Author response image 1). It could also be mediated by downstream pathways of the circadian regulatory network. Notably, the Nr1d1/Nr1d2 nuclear receptor family members have been shown to negatively regulate macrophage inflammatory gene expression (Lam et al., 2013). The induction of Nr1d2 by M1 was almost completely abolished in M-BKO macrophages (Figure 1D). In concert, the expression of *Cx3cr1*, a direct target suppressed by Nr1d1/Nr1d2 (Lamet al., 2013), was higher in M1 activated M-BKO macrophages, compared to WT cells (Figure 5—source data 1). This suppressive effect of Nr1d1/Nr1d2 may dampen the inflammatory damage to mitochondrial function at the initial stage of M1 activation. We have discussed these potential mechanisms in the revised Discussion section.

3) The findings on tumor associated macrophages (TAMs) need to be extended. Specifically, do TAMs derived from tumors display a similar regulation of Bmal1-mitochondrial capacity and glucose utilization as bone marrow-derived macrophages exposed to tumor derived conditioned medium?

We agree it’s important that the proposed mechanism could be demonstrated in TAMs derived from tumors. Unfortunately, it is technically challenging to obtain sufficient TAMs from mouse tumor models for functional assays, such as mitochondrial respiration and glucose utilization. Using flow cytometry, we were able to stain F4/80+ cells from tumor lysate with 2NBDG, a fluorescent glucose analog, and Mitosox red to assess glucose uptake and mROS levels, respectively. As showed in the revised Figure 6—figure supplement 1A-1C, M-BKO TAMs from primary tumors also exhibited increased glucose uptake and mROS production, compared to WT TAMs.

4) The data presented in Figure 1—figure supplement 1 indicate that the clock program downstream of Bmal is activated upon M1 induction. This raises a number of questions about the relationship between Bmal role in metabolic regulation and its well-known activity as a molecular clock. Discussing these findings in light of existing literature would be helpful for the audience to understand their possible implications in terms of circadian regulation of immunometabolism.

We agree with the reviewers that Bmal1 is best known for its role as a master circadian regulator and our RNA-seq data indicate the clock program downstream of Bmal1 is activated by M1 activation. Studies have shown that certain components of the molecular clock are capably of suppressing the expression of inflammatory genes, including Nr1d1/Nr1d2 nuclear receptor family members discussed in point 2.2. Bmal1 has also been shown to regulate diurnal monocyte trafficking that appears to play an important role in immune response against pathogens and in limiting inflammatory damage (Nguyen et al., 2013). We think that as a circadian regulator, Bmal1 could modulate inflammatory gene expression through downstream effectors, such as Nr1d1/Nr1d2. As a metabolic regulator, Bmal1-mediated energetic control may provide energy needed for diurnal monocyte trafficking/migration. In the context of infection or LPS/M1 activation, Bmal1 may regulate the timing of glycolytic to oxidative metabolism transition that dictates the extent of Hif-1α activation and the associated inflammatory response/damage. As suggested by the reviewers, we have incorporated these points into the second paragraph of the Discussion.